# Improving Generalization of Meta Reinforcement Learning via Explanation

## Abstract

Meta reinforcement learning learns a meta-prior (e.g., meta-policy) from a set of training tasks, such that the learned meta-prior can efficiently adapt to all the tasks in a task distribution. However, it has been observed in literature that the learned meta-prior usually has imbalanced generalization, i.e., it adapts well to some tasks but adapts poorly to some other tasks. This paper aims to explain why certain tasks are poorly adapted and, more importantly, use this explanation to improve generalization. Our methodology has two parts. The first part identifies "critical" training tasks that are most important to achieve good performance on those poorly-adapted tasks. An explanation of the poor generalization is that the meta-prior does not pay enough attention to the critical training tasks. To improve generalization, the second part formulates a bi-level optimization problem where the upper level learns how to augment the critical training tasks such that the meta-prior can pay more attention to the critical tasks, and the lower level computes the meta-prior distribution corresponding to the current augmentation. We propose an algorithm to solve the bi-level optimization problem and theoretically guarantee that (1) the algorithm converges at the rate of $O(1/\sqrt{K})$, (2) the learned augmentation makes the meta-prior focus more on the critical training tasks, and (3) the generalization improves after the task augmentation. We use two real-world experiments and three MuJoCo experiments to show that our algorithm improves the generalization and outperforms state-of-the-art baselines.

## 1 Introduction

Meta reinforcement learning (Meta-RL) aims to learn a meta-prior from a set of training tasks where each training task is an RL problem and is drawn from an implicit task distribution. The learned meta-prior is expected to adapt well (i.e., achieve high cumulative reward after adaptation) to every task in this task distribution (Beck et al., 2023). However, it has been observed (Dhillon et al., 2019; Nguyen et al., 2021; Yu et al., 2020) that the learned meta-prior usually does not adapt well to all the tasks in the task distribution, i.e., it adapts well to some tasks but adapts poorly to some other tasks. This paper proposes the first method that uses explainable meta-RL to improve generalization. Our methodology has two parts. The first part explains why certain tasks are poorly adapted by identifying the mistakes made by the meta-prior. The second part uses the explanation in the first part to help correct the mistakes and thus improve generalization.

The first part explains why certain tasks are poorly adapted from the perspective of training tasks. One reason (Nguyen et al., 2023) of this poor generalization phenomenon is that the meta-prior is learned by minimizing the average loss of all the training tasks, implicitly treating all the training tasks as equally important. However, many studies have shown that (Thrun & O'Sullivan, 1996; Nguyen et al., 2023; Zamir et al., 2018; Achille et al., 2019; Nguyen et al., 2021) the tasks are not equally important, instead, paying attention to certain *important* tasks can facilitate the generalization over the whole task distribution. Treating all the training tasks as equally important can potentially hinder the meta-prior from paying enough attention to some important tasks, and thus lead to poor generalization. Inspired by the idea of identifying critical states that are most influential to the cumulative reward as an explanation in explainable RL (Guo et al., 2021b; Cheng et al., 2024), we aim to identify the training tasks that are most important to achieve good performance on those poorly adapted tasks. We refer to these training tasks as "critical tasks". Our explanation for the poor generalization is that the meta-prior does not pay enough attention to the critical tasks.

The second part aims to improve generalization by encouraging the meta-prior to pay more attention to the critical training tasks. Since the critical tasks are the most important tasks to achieve high cumulative reward on those poorly-adapted tasks, paying more attention to the critical tasks results in a new meta-prior that generalizes better to those originally poorly-adapted tasks. Since this new meta-prior generalizes well to additional tasks compared to the original meta-prior, the generalization over the whole task distribution is likely to improve. To encourage the meta-prior to pay more attention to the critical tasks, we propose to augment the critical tasks by generating augmented data and train the meta-prior over the augmented data. The augmented data increases the diversity of the original data and contains additional information. Therefore, it is expected that the meta-prior trained on the augmented data stores more information of the critical task and thus pays more attention to the critical tasks. Some recent works augment data to facilitate generalization in RL (Wang et al., 2020) and meta-learning (Rajendran et al., 2020; Yao et al., 2021). However, they use a pre-defined rule to augment the data or tasks. While the pre-defined rule may provide a feasible augmentation, it is not the optimal augmentation, i.e., the augmentation that enables the learned model to best pay attention to the critical tasks. This paper formulates a bi-level optimization problem where the upper level learns how to best augment the critical tasks and the lower level computes the meta-prior distribution corresponding to the current augmentation. In the upper level, we use an information theoretic metric to quantify the information of the critical tasks stored in the meta-prior. Intuitively, the more information of the critical tasks stored in the meta-prior, the more attention the meta-prior pays to the critical tasks. Therefore, we aim to learn an augmentation method to maximize this stored information. The difficulty of the upper-level optimization is that we need to compute a distribution of the meta-prior. Therefore, the lower level formulates a distributional optimization problem where a meta-prior distribution, instead of a single meta-prior, corresponding to the current augmentation is learned. We summarize our contributions as follows.

**Contribution statement**. This paper proposes the first method that uses explainable meta-RL to improve generalization of meta-RL. Our contributions are threefold:

First, we propose the first explainable meta-RL method. Our method explains why the learned meta-prior adapts poorly to certain tasks by identifying the critical training tasks where the meta-prior does not pay enough attention.

Second, we formalize the problem of utilizing the explanation to improve generalization as a bi-level optimization problem where the upper level learns how to augment the critical tasks such that the meta-prior can best pay attention to the critical tasks, and the lower level computes the meta-prior distribution corresponding to the current augmentation. We propose a novel algorithm to solve the bi-level optimization problem.

Third, we theoretically guarantee that (1) our algorithm converges at the rate of $O(1/\sqrt{K})$, (2) the learned augmentation makes the meta-prior focus more on the critical tasks, and (3) the generalization improves after the task augmentation. We use two real-world experiments and three MuJoCo experiments to empirically show that our algorithm can improve generalization of meta reinforcement learning and outperform state-of-the-art baselines.

## 2 RELATED WORKS

This section discusses related works. Note that there is no previous work on explainable meta-RL. We introduce works in the following three related areas: explainable RL, explainable meta-learning, and meta-learning generalization improvement. We also discuss our distinctions from the literature.

**Explainable reinforcement learning**. While it lacks research works on explainable meta-RL, explainable RL (XRL) has been extensively studied to explain the decision-making of the RL agents, including learning an interpretable policy (Bastani et al., 2018; Bewley & Lawry, 2021; Verma et al., 2018), pinpointing regions in the observations that are critical for choosing certain actions (Atrey et al., 2019; Guo et al., 2021a; Puri et al., 2019), and reward decomposition (Juozapaitis et al., 2019; Lin et al., 2020a; Septon et al., 2023). The most relevant XRL method to our explanation is to identify the critical states that are influential to the cumulative reward as an explanation (Guo et al., 2021b; Cheng et al., 2024; Amir & Amir, 2018) where they respectively use an RNN, masks, and a self-proposed rule to find critical states. In contrast, we formulate a bi-level optimization problem to learn a weight vector that indicates critical tasks.

**Explainable meta-learning**. There are three works on explainable meta-learning where (Woźnica & Biecek, 2021) proposes to learn important features that lead to a specific meta model decision using Friedman's H-statistic (Friedman & Popescu, 2008), and (Shao et al., 2022; 2023) use structural causal model to model the causal relations between the features and the model decision. While these works explain why a decision is made, we explain why certain tasks are poorly adapted.

**Meta-learning generalization improvement**. There are three major ways to improve meta-learning generalization: task weighting, regularization, and meta-augmentation. Task weighting (Cai et al., 2020; Yao et al., 2021; Nguyen et al., 2023) proposes to re-weight the training tasks or reshape the training task distribution to improve generalization. However, (Cai et al., 2020; Yao et al., 2021) require an additional target task set to guide how to weight the training tasks or reshape the training task distribution, and thus the learned meta-prior can be biased towards the target task set and may not adapt well to other tasks. Regularization-based methods are also used to improve generalization where (Wang et al., 2023) proposes to add ordinary regularization to the upper level and inverted regularization to the lower level, and (Yin et al., 2019) imposes regularization to prevent memorization overfitting. The most relevant technique to our paper is meta-augmentation which augments the data and train on the augmented data to improve generalization. In specific, (Rajendran et al., 2020) proposes to add noise to the data and (Yao et al., 2021) proposes to mix data and shuffle the channels in the hidden layers. The augmentation method has also been used in RL (Wang et al., 2020) to improve generalization. These augmentation methods use pre-defined rules to provide feasible augmentation. In contrast, our paper aims to learn how to best augment the critical tasks.

## 3 PRELIMINARIES

**Reinforcement learning**. An RL task $\mathcal{T}_i$ is based on a Markov decision process (MDP) $\mathcal{M}_i = (\mathcal{S}, \mathcal{A}, \gamma, P_i, \nu_i, r_i)$ which includes a state set $\mathcal{S}$, an action set $\mathcal{A}$, a discount factor $\gamma \in (0, 1)$, a state transition function $P_i(\cdot|\cdot, \cdot)$, an initial state distribution $\nu_i(\cdot)$, and a reward function $r_i(\cdot, \cdot)$. Reinforcement learning aims to learn a policy $\pi_\varphi$ (parameterized by $\varphi$) to maximize the cumulative reward, i.e., $\max_\varphi E^{\pi_\varphi}[\sum_{t=0}^{\infty} \gamma^t r_i(s_t, a_t)|s_0 \sim \nu_i]$. The policy gradient (Sutton et al., 1999) is $E_{(s,a) \sim \rho^{\pi_\varphi}}[\nabla_\varphi \log \pi_\varphi(a|s) A_i^{\pi_\varphi}(s, a)]$ where $A_i^\pi$ is the advantage function under the reward $r_i$ and policy $\pi$, $\rho^\pi(s, a) \triangleq E^\pi[\sum_{t=0}^{\infty} \gamma^t \mathbb{1}\{s_t = s, a_t = a\}|s_0 \sim \nu_i]$ is the stationary state-action distribution under the policy $\pi$, and $\mathbb{1}\{\cdot\}$ is the indicator function. Based on the policy gradient, we can formulate a surrogate objective for RL (Wang et al., 2020): $J_i(\pi) \triangleq E_{(s,a) \sim \rho^\pi}[\log \pi(a|s) A_i^\pi(s, a)]$. Here, we omit the policy parameter $\varphi$.

**Meta reinforcement learning**. Meta-RL aims to efficiently solve multiple RL tasks by learning a meta-prior. The meta-prior is learned from a group of $N^{\mathrm{tr}}$ training tasks $\{\mathcal{T}_i^{\mathrm{tr}}\}_{i=1}^{N^{\mathrm{tr}}}$ sampled from an implicit task distribution $P(\mathcal{T})$. It is typically assumed (Beck et al., 2023) that different tasks share $(\mathcal{S}, \mathcal{A}, \gamma)$ but may have different $(P_i^{\mathrm{tr}}, \nu_i^{\mathrm{tr}}, r_i^{\mathrm{tr}})$. Here, the superscript "tr" means that these components belong to training tasks. Later on, we will use different superscripts to represent different kinds of tasks. Current mainstream meta-RL works (Beck et al., 2023; Finn et al., 2017; Fallah et al., 2021; Xu et al., 2018; Liu et al., 2019) learn a meta-policy $\pi_\theta$ (as the meta-prior) from the training tasks and have the following bi-level structure:

$$\max_\theta \ L(\theta, \{\mathcal{T}_i^{\mathrm{tr}}\}_{i=1}^{N^{\mathrm{tr}}}) = \frac{1}{N^{\mathrm{tr}}} \sum_{i=1}^{N^{\mathrm{tr}}} J_i^{\mathrm{tr}}(\pi_i^{\mathrm{tr}}(\theta)), \quad \text{s.t. } \pi_i^{\mathrm{tr}}(\theta) = Alg(\pi_\theta, \mathcal{T}_i^{\mathrm{tr}}), \quad (1)$$

where the upper level aims to learn a meta-policy $\pi_\theta$ such that the corresponding task-specific adaptation $\pi_i^{\mathrm{tr}}(\theta)$ can maximize the cumulative reward $J_i^{\mathrm{tr}}(\pi_i^{\mathrm{tr}}(\theta))$ on each training task $\mathcal{T}_i^{\mathrm{tr}}$, and the lower level computes the task-specific adaptation $\pi_i^{\mathrm{tr}}(\theta)$ given the meta-parameter $\theta$. Different meta-learning methods use different algorithms to compute the task-specific adaptation $\pi_i^{\mathrm{tr}}(\theta)$. Here, we use $Alg(\pi_\theta, \mathcal{T}_i^{\mathrm{tr}})$ to generally represent an algorithm that computes the task-specific adaptation.

To evaluate the generalization of the meta-policy $\pi_\theta$ over the task distribution $P(\mathcal{T})$, people usually sample some validation tasks where each validation task is drawn from $P(\mathcal{T})$, and use the task-specific adaptation of each validation task to test the performance on each validation task. However, it has been empirically observed (Yu et al., 2020) that only a portion of the adapted policies perform well on the corresponding validation tasks while some adapted policies perform poorly on

the corresponding validation tasks. We pick the top $N^{\text{poor}}$ poorly-adapted validation tasks and use $\{\mathcal{T}_i^{\text{poor}}\}_{i=1}^{N^{\text{poor}}}$ to represent the set of these top $N^{\text{poor}}$ poorly-adapted validation tasks.

This paper aims to improve the generalization of the meta-policy $\pi_\theta$ via two steps. The first step aims to explain why $\pi_\theta$ adapts poorly to $\{\mathcal{T}_i^{\text{poor}}\}_{i=1}^{N^{\text{poor}}}$. The second step aims to use the explanation in the first step to improve the generalization over $P(\mathcal{T})$.

## 4  THE EXPLANATION

This section explains why the meta-policy $\pi_\theta$ does not adapt well to $\{\mathcal{T}_i^{\text{poor}}\}_{i=1}^{N^{\text{poor}}}$ from the perspective of the training tasks. In specific, the meta-policy $\pi_\theta$ is learned by minimizing the average loss of the training tasks, implicitly treating all the training tasks as equally important. However, many studies (Thrun & O'Sullivan, 1996; Zamir et al., 2018; Achille et al., 2019; Nguyen et al., 2021) have shown that the tasks are not equally important, and learning from certain *important* tasks can facilitate the generalization performance. Treating all the training tasks as equally important can potentially hinder the meta-prior from paying enough attention to some important tasks. Therefore, an explanation of why $\pi_\theta$ does not adapt well to $\{\mathcal{T}_i^{\text{poor}}\}_{i=1}^{N^{\text{poor}}}$ is that $\pi_\theta$ does not pay enough attention to some training tasks that are most important to achieve high cumulative reward on $\{\mathcal{T}_i^{\text{poor}}\}_{i=1}^{N^{\text{poor}}}$. We refer to these training tasks as "critical tasks" and aim to identify the top $N^{\text{cri}}$ critical training tasks as an explanation.

For this purpose, we propose to learn an importance vector $\omega \in \mathbb{R}^{N^{\text{tr}}}$ where each dimension $\omega_i$ captures the importance of the corresponding training task $\mathcal{T}_i^{\text{tr}}$ in terms of achieving high cumulative reward on $\{\mathcal{T}_i^{\text{poor}}\}_{i=1}^{N^{\text{poor}}}$. In specific, we propose to solve a bi-level optimization problem:

$$\max_{\omega} \ L(\theta^*(\omega), \{\mathcal{T}_i^{\text{poor}}\}_{i=1}^{N^{\text{poor}}}) \quad \text{s.t. } \theta^*(\omega) = \arg\max_{\theta} \sum_{i=1}^{N^{\text{tr}}} \omega_i J_i^{\text{tr}}(\pi_i^{\text{tr}}(\theta)), \tag{2}$$

where the upper level aims to learn how to weight each training task such that the corresponding weighted meta-policy $\pi_{\theta^*(\omega)}$ can adapt to $\{\mathcal{T}_i^{\text{poor}}\}_{i=1}^{N^{\text{poor}}}$ with maximum cumulative reward, and the lower level computes the weighted meta-policy $\pi_{\theta^*(\omega)}$ corresponding to the current weight $\omega$. We include the algorithm to solve the problem (2) in Appendix A.

We use $\omega^*$ to denote an optimal solution of problem (2). A higher $\omega_i^*$ means that the weighted meta-policy $\pi_{\theta^*(\omega^*)}$ should weight the training task $\mathcal{T}_i^{\text{tr}}$ more in order to adapt to $\{\mathcal{T}_i^{\text{poor}}\}_{i=1}^{N^{\text{poor}}}$ with high cumulative reward, and thus the training task $\mathcal{T}_i^{\text{tr}}$ is more important in terms of achieving high cumulative reward on $\{\mathcal{T}_i^{\text{poor}}\}_{i=1}^{N^{\text{poor}}}$. Therefore, the top $N^{\text{cri}}$ training tasks with the highest weight values are the top $N^{\text{cri}}$ critical tasks we aim to identify. We use $\{\mathcal{T}_i^{\text{cri}}\}_{i=1}^{N^{\text{cri}}}$ to denote these $N^{\text{cri}}$ critical training tasks.

**Remark 1** (**The weighted meta-policy $\pi_{\theta^*(\omega^*)}$ cannot be used to improve generalization**). *Note that $\pi_{\theta^*(\omega^*)}$ only improves generalization to the poorly-adapted tasks $\{\mathcal{T}_i^{\text{poor}}\}_{i=1}^{N^{\text{poor}}}$, but can compromise the performance on the non-critical tasks (i.e., the other training tasks that are not the critical tasks) and thus potentially compromise the generalization to the tasks similar to the non-critical tasks. The reason is that $\pi_{\theta^*(\omega^*)}$ is trained to solve a biased problem (i.e., the lower-level problem in (2)) where the critical tasks are assigned with larger weights and the non-critical tasks are assigned with smaller weights. This bias enables $\pi_{\theta^*(\omega^*)}$ to generalize better to the originally poorly-adapted tasks $\{\mathcal{T}_i^{\text{poor}}\}_{i=1}^{N^{\text{poor}}}$. However, since $\pi_{\theta^*(\omega^*)}$ is biased towards optimizing the performance on the critical tasks, the performance on the non-critical tasks becomes secondary and can be compromised, especially when the non-critical tasks are very different from the critical tasks. Therefore, this bias can potentially hinder the meta-policy from generalizing well to tasks similar to the non-critical tasks.*

**Remark 2** (**Improving generalization without introducing new training tasks**). *A simple way to improve generalization is to include more training tasks, especially the tasks similar to the poorly-adapted tasks $\{\mathcal{T}_i^{\text{poor}}\}_{i=1}^{N^{\text{poor}}}$. However, this paper aims to improve generalization without introducing new training tasks. Moreover, our method is complementary to the method of introducing new training tasks because one can both introduce new training tasks and use our method to improve generalization.*

## 5 THE IMPROVEMENT

This section uses the explanation (i.e., the critical tasks $\{\mathcal{T}_i^{\mathrm{cri}}\}_{i=1}^{N^{\mathrm{cri}}}$) in Section 4 to improve generalization by encouraging the meta-policy to focus more on the critical tasks. Since the critical tasks are the most important tasks to achieve high cumulative reward on the poorly-adapted tasks $\{\mathcal{T}_i^{\mathrm{poor}}\}_{i=1}^{N^{\mathrm{poor}}}$, paying more attention to the critical tasks results in a new meta-policy that generalizes better to the originally poorly-adapted tasks. Since this new meta-policy generalizes well to additional tasks compared to the original meta-policy (i.e., the one without paying more attention to the critical tasks), the generalization over the whole task distribution is likely to improve. The *challenge* is to design a method that enables the meta-policy to focus more on the critical tasks.

A straightforward method to focus more on the critical tasks is to assign larger weights to the critical tasks and train a meta-policy over the weighted training tasks. However, as mentioned in Remark 1, while this weighting method can improve generalization to the originally poorly-adapted tasks, it makes the meta-policy biased towards the critical tasks and can compromise the performance on the non-critical tasks. Therefore, this bias may potentially hinder the meta-policy from generalizing well to tasks similar to the non-critical tasks.

To address this issue, we propose to focus more on the critical tasks by augmenting the critical training tasks. We generate augmented data for the critical tasks where the augmented data increases the diversity of the data and thus contains additional information. We train the meta-policy over the non-critical training tasks and the augmented critical training tasks. Since the meta-policy is trained on the augmented data that contains additional information of the critical tasks, it is expected that the meta-policy stores more task information of the critical tasks and thus pays more attention to the critical tasks. Compared to directly assigning larger weights to the critical tasks, the benefit of the proposed task augmentation is that it does not compromise the performance on the non-critical tasks because it does not introduce bias towards the critical tasks and the task information of the non-critical training tasks stored in the meta-policy remains unchanged (proved in Appendix B).

This section has three parts. The first part formulates a bi-level optimization problem to learn how to best augment the critical tasks such that the meta-policy can focus more on the critical tasks. The second part proposes a novel algorithm to solve the bi-level optimization problem. The third part includes the theoretical analysis which proves that (1) the algorithm converges at the rate of $O(1/\sqrt{K})$, (2) the learned augmentation makes the meta-prior focus more on the critical tasks, and (3) the generalization improves after the task augmentation.

### 5.1 PROBLEM FORMULATION

This part formulates a bi-level optimization problem where the upper level aims to learn how to augment the critical tasks such that the meta-policy can best pay attention to the critical tasks, and the lower level computes the meta-parameter distribution corresponding to the current augmentation.

We use data mixture to augment the critical tasks where data mixture can increase the diversity of the original data and thus contain additional information (Yao et al., 2021; Wang et al., 2020). Recall from Section 3 that we can formulate a surrogate RL objective for the critical task $\mathcal{T}_i^{\mathrm{cri}}$: $J_i^{\mathrm{cri}}(\pi) = E_{(s,a)\sim\rho^\pi}[\log \pi(a|s)A_i^\pi(s,a)]$. Data mixture (Wang et al., 2020) proposes to mix any two data points $(s_j, a_j, A_i^\pi(s_j, a_j))$ and $(s_{j'}, a_{j'}, A_i^\pi(s_{j'}, a_{j'}))$ to generate augmented data $(\bar{s}_{jj'}, \bar{a}_{jj'}, \bar{A}_{jj'})$ where $\bar{s}_{jj'} = \lambda_i s_j + (1 - \lambda_i)s_{j'}$, $\bar{A}_{jj'} = \lambda_i A_i^\pi(s_j, a_j) + (1 - \lambda_i)A_i^\pi(s_{j'}, a_{j'})$, $\bar{a}_{jj'} = a_j$ if $\lambda_i \geq 0.5$ and $\bar{a}_{jj'} = a_{j'}$ if $\lambda_i < 0.5$, and the mixture coefficient $\lambda_i \in [0, 1]$ of the critical task $\mathcal{T}_i^{\mathrm{cri}}$ is a random variable that is drawn from a distribution $P(\lambda)$. For a specific $\lambda_i$, the data mixture will lead to an augmented stationary state-action distribution $\bar{\rho}^{\pi,\lambda_i}(\bar{s}_{jj'}, \bar{a}_{jj'})$ whose expression is in Appendix C. Therefore, we have an augmented task $\bar{\mathcal{T}}_i^{\mathrm{cri}}(\lambda_i)$ and its surrogate RL objective is $\bar{J}_i^{\mathrm{cri}}(\pi, \lambda_i) \triangleq E_{(\bar{s}_{jj'}, \bar{a}_{jj'})\sim\bar{\rho}^{\pi,\lambda_i}}[\log \pi(\bar{a}_{jj'}|\bar{s}_{jj'})\bar{A}_{jj'}]$. With the augmented critical tasks, the meta-objective (i.e., the upper-level objective) in (1) becomes:

$$L(\theta, \{\bar{\mathcal{T}}_i^{\mathrm{cri}}(\lambda_i)\}_{i=1}^{N^{\mathrm{cri}}}, \{\mathcal{T}_i^{\mathrm{tr}}\}_{i=1}^{N^{\mathrm{tr}}-N^{\mathrm{cri}}}) \triangleq \frac{1}{N^{\mathrm{tr}}}\Big[\sum_{i=1}^{N^{\mathrm{cri}}} \bar{J}_i^{\mathrm{cri}}(\pi_i^{\mathrm{cri}}(\theta), \lambda_i) + \sum_{i=1}^{N^{\mathrm{tr}}-N^{\mathrm{cri}}} J_i^{\mathrm{tr}}(\pi_i^{\mathrm{tr}}(\theta))\Big]. \quad (3)$$

Compared to the original meta-objective in (1), the new objective (3) replaces the original critical tasks $\{\mathcal{T}_i^{\mathrm{cri}}\}_{i=1}^{N^{\mathrm{cri}}}$ with the augmented critical tasks $\{\bar{\mathcal{T}}_i^{\mathrm{cri}}(\lambda_i)\}_{i=1}^{N^{\mathrm{cri}}}$. Since $\lambda_i$ is a random variable, the

corresponding augmented task $\bar{\mathcal{T}}_i^{\mathrm{cri}}(\lambda_i)$ is also a random variable. In the following context, we use the notation $\bar{\mathcal{T}}_i^{\mathrm{cri}}(\lambda_i \sim P(\lambda))$ to highlight that the augmented task is a random variable.

To mathematically reason about whether the meta-parameter pays more attention to the critical tasks after task augmentation, we use the following information theoretic metric:

**Definition 1.** *We say that the meta-parameter pays more attention to the critical tasks after task augmentation if $I(\theta; \{\bar{\mathcal{T}}_i^{\mathrm{cri}}(\lambda_i \sim P(\lambda))\}_{i=1}^{N^{\mathrm{cri}}} | \{\mathcal{T}_i^{\mathrm{cri}}\}_{i=1}^{N^{\mathrm{cri}}}) > 0$ where $I(\theta; \{\bar{\mathcal{T}}_i^{\mathrm{cri}}(\lambda_i \sim P(\lambda))\}_{i=1}^{N^{\mathrm{cri}}} | \{\mathcal{T}_i^{\mathrm{cri}}\}_{i=1}^{N^{\mathrm{cri}}})$ is the conditional mutual information between the meta-parameter $\theta$ and the augmented critical tasks $\{\bar{\mathcal{T}}_i^{\mathrm{cri}}(\lambda_i \sim P(\lambda))\}_{i=1}^{N^{\mathrm{cri}}}$, given the original critical tasks $\{\mathcal{T}_i^{\mathrm{cri}}\}_{i=1}^{N^{\mathrm{cri}}}$.*

In information theory (Wyner, 1978; Yao et al., 2021), the conditional mutual information quantifies the difference between the information that $\theta$ and $\{\bar{\mathcal{T}}_i^{\mathrm{cri}}(\lambda_i \sim P(\lambda))\}_{i=1}^{N^{\mathrm{cri}}}$ share and the information that $\theta$ and $\{\mathcal{T}_i^{\mathrm{cri}}\}_{i=1}^{N^{\mathrm{cri}}}$ share. In other words, the conditional mutual information quantifies the amount of additional information stored in $\theta$ by additionally knowing $\{\bar{\mathcal{T}}_i^{\mathrm{cri}}(\lambda_i \sim P(\lambda))\}_{i=1}^{N^{\mathrm{cri}}}$ given that $\{\mathcal{T}_i^{\mathrm{cri}}\}_{i=1}^{N^{\mathrm{cri}}}$ is already known. Intuitively, $I(\theta; \{\bar{\mathcal{T}}_i^{\mathrm{cri}}(\lambda_i \sim P(\lambda))\}_{i=1}^{N^{\mathrm{cri}}} | \{\mathcal{T}_i^{\mathrm{cri}}\}_{i=1}^{N^{\mathrm{cri}}}) > 0$ means that the task information of the critical tasks stored in the meta-parameter $\theta$ increases after we augment $\{\mathcal{T}_i^{\mathrm{cri}}\}_{i=1}^{N^{\mathrm{cri}}}$ to $\{\bar{\mathcal{T}}_i^{\mathrm{cri}}(\lambda_i \sim P(\lambda))\}_{i=1}^{N^{\mathrm{cri}}}$. Since the task information of the critical tasks stored in $\theta$ increases, it means that the meta-parameter $\theta$ pays more attention to the critical tasks.

We aim to augment the critical tasks such that $I(\theta; \{\bar{\mathcal{T}}_i^{\mathrm{cri}}(\lambda_i \sim P(\lambda))\}_{i=1}^{N^{\mathrm{cri}}} | \{\mathcal{T}_i^{\mathrm{cri}}\}_{i=1}^{N^{\mathrm{cri}}}) > 0$. However, the aforementioned data mixture works (Yao et al., 2021; Wang et al., 2020) use a predetermined distribution $P(\lambda)$ of $\lambda_i$ to mix the data. While (Yao et al., 2021) shows that the predetermined distribution $P(\lambda)$ is a feasible augmentation to increase the task information stored in the meta-parameter, this distribution is not guaranteed to be an optimal augmentation, i.e., the one that can maximally increase the task information stored in the meta-parameter. We aim to learn how to best augment the critical tasks $\{\mathcal{T}_i^{\mathrm{cri}}\}_{i=1}^{N^{\mathrm{cri}}}$ by optimizing for the distribution $P(\lambda)$ such that the conditional mutual information can be maximized. In specific, we use a parameterized distribution $P_{\phi_\lambda}(\lambda)$ with parameter $\phi_\lambda$ to model the distribution of $\lambda$. We aim to optimize the distribution parameter $\phi_\lambda$ to maximize the conditional mutual information. The expression of the conditional mutual information is:

$$I(\theta; \{\bar{\mathcal{T}}_i^{\mathrm{cri}}(\lambda_i \sim P_{\phi_\lambda}(\lambda))\}_{i=1}^{N^{\mathrm{cri}}} | \{\mathcal{T}_i^{\mathrm{cri}}\}_{i=1}^{N^{\mathrm{cri}}}),$$

$$= E_{\lambda_i \in [0,1], \lambda_i \sim P_{\phi_\lambda}(\lambda), \theta \sim P^*(\cdot | \{\bar{\mathcal{T}}_i^{\mathrm{cri}}(\lambda_i)\}_{i=1}^{N^{\mathrm{cri}}})} \left[ \log \frac{P^*(\theta | \{\bar{\mathcal{T}}_i^{\mathrm{cri}}(\lambda_i)\}_{i=1}^{N^{\mathrm{cri}}})}{P^*(\theta | \{\mathcal{T}_i^{\mathrm{cri}}\}_{i=1}^{N^{\mathrm{cri}}})} \right], \quad (4)$$

where the derivation is in Appendix D, $P^*(\cdot | \{\bar{\mathcal{T}}_i^{\mathrm{cri}}(\lambda_i)\}_{i=1}^{N^{\mathrm{cri}}})$ is the posterior distribution of the meta-parameter $\theta$ given the augmented critical tasks $\{\bar{\mathcal{T}}_i^{\mathrm{cri}}(\lambda_i)\}_{i=1}^{N^{\mathrm{cri}}}$, and $P^*(\cdot | \{\mathcal{T}_i^{\mathrm{cri}}\}_{i=1}^{N^{\mathrm{cri}}})$ is the posterior distribution of $\theta$ given the original critical tasks $\{\mathcal{T}_i^{\mathrm{cri}}\}_{i=1}^{N^{\mathrm{cri}}}$. Note that the posterior distributions of $\theta$ should also depend on the non-critical training tasks $\{\mathcal{T}_i^{\mathrm{tr}}\}_{i=1}^{N^{\mathrm{tr}} - N^{\mathrm{cri}}}$, here, we omit the dependence of the non-critical tasks because the non-critical tasks do not change after task augmentation.

To maximize the conditional mutual information (4), we need to compute the posterior distributions $P^*(\theta | \{\mathcal{T}_i^{\mathrm{cri}}\}_{i=1}^{N^{\mathrm{cri}}})$ and $P^*(\theta | \{\bar{\mathcal{T}}_i^{\mathrm{cri}}(\lambda_i)\}_{i=1}^{N^{\mathrm{cri}}})$. Therefore, analogous to (Achille & Soatto, 2018; Yin et al., 2019), we treat $\theta$ as a random variable where the randomness comes from the training stochasticity. Mathematically, the posterior distributions are:

$$P^*(\cdot | \{\bar{\mathcal{T}}_i^{\mathrm{cri}}(\lambda_i)\}_{i=1}^{N^{\mathrm{cri}}}) = \arg\max_\phi E_{p_\phi(\theta)} \left[ L(\theta, \{\bar{\mathcal{T}}_i^{\mathrm{cri}}(\lambda_i)\}_{i=1}^{N^{\mathrm{cri}}}, \{\mathcal{T}_i^{\mathrm{tr}}\}_{i=1}^{N^{\mathrm{tr}} - N^{\mathrm{cri}}}) \right],$$

$$P^*(\cdot | \{\mathcal{T}_i^{\mathrm{cri}}\}_{i=1}^{N^{\mathrm{cri}}}) = E_{\lambda_i \in [0,1], \lambda_i \sim P_{\phi_\lambda}(\lambda)} \left[ P^*(\cdot | \{\bar{\mathcal{T}}_i^{\mathrm{cri}}(\lambda_i)\}_{i=1}^{N^{\mathrm{cri}}}) \right] \quad (5)$$

where $P_\phi(\theta)$ is a distribution of $\theta$ parameterized by $\phi$. Instead of learning a single meta-parameter $\theta$, problem (5) aims to learn a distribution of $\theta$ that can maximize the meta-objective (3). This idea of optimizing a distribution is widely adopted in meta-learning (Yin et al., 2019) and RL (Liu et al., 2017; Salimans et al., 2017) when the stochasticity of the learned parameter is of interest. By combining (4) and (5), we reach the final bi-level optimization problem:

$$\max_{\phi_\lambda} I(\theta; \{\bar{\mathcal{T}}_i^{\mathrm{cri}}(\lambda_i \sim P_{\phi_\lambda}(\lambda))\}_{i=1}^{N^{\mathrm{cri}}} | \{\mathcal{T}_i^{\mathrm{cri}}\}_{i=1}^{N^{\mathrm{cri}}}), \quad \text{s.t.} \quad \text{Problem (5)}, \quad (6)$$

where the upper-level problem in (6) learns a distribution $P_{\phi_\lambda}(\lambda)$ of the mixture coefficients $\{\lambda_i\}_{i=1}^{N^{\text{cri}}}$ to maximize the conditional mutual information (4) (i.e., maximally increase the additional information of the critical tasks stored in the meta-parameter), and the lower level (i.e., problem (5)) computes the posterior distribution $P^*(\theta|\{\bar{\mathcal{T}}_i^{\text{cri}}(\lambda_i)\}_{i=1}^{N^{\text{cri}}})$ corresponding to the current mixture coefficients $\{\lambda_i\}_{i=1}^{N^{\text{cri}}}$ and the posterior distribution $P^*(\theta|\{\mathcal{T}_i^{\text{cri}}\}_{i=1}^{N^{\text{cri}}})$ given the original critical tasks.

## 5.2 ALGORITHM

In this section, we develop an algorithm to improve the generalization of meta-RL. We first identify the critical tasks as the explanation (line 1 in Algorithm 1). With the identified critical tasks, we encourage the meta-parameter $\theta$ to focus more on the critical tasks by solving the problem (6). At each iteration $k$, we first solve the lower-level problem (5) in line 3. In specific, we first sample $N^{\bar{\zeta}}$ sets of mixture coefficients $\{\{\lambda_{i,k}^{\bar{\zeta}_j}\}_{i=1}^{N^{\text{cri}}}\}_{j=1}^{N^{\bar{\zeta}}}$ from $P_{\phi_\lambda,k}(\lambda)$ and project each $\lambda_{i,k}^{\bar{\zeta}_j}$ to $[0,1]$, and then compute $N^{\bar{\zeta}}$ posterior distributions $\{P^*(\cdot|\{\bar{\mathcal{T}}_i^{\text{cri}}(\lambda_{i,k}^{\bar{\zeta}_j})\}_{i=1}^{N^{\text{cri}}})\}_{\bar{\zeta}_j=1}^{N^{\bar{\zeta}}}$ where each posterior distribution $P^*(\cdot|\{\bar{\mathcal{T}}_i^{\text{cri}}(\lambda_{i,k}^{\bar{\zeta}_j})\}_{i=1}^{N^{\text{cri}}})$ corresponds to each set of mixture coefficients $\{\lambda_{i,k}^{\bar{\zeta}_j}\}_{i=1}^{N^{\text{cri}}}$. We use these $N^{\bar{\zeta}}$ posterior distributions $\{P^*(\cdot|\{\bar{\mathcal{T}}_i^{\text{cri}}(\lambda_{i,k}^{\bar{\zeta}_j})\}_{i=1}^{N^{\text{cri}}})\}_{\bar{\zeta}_j=1}^{N^{\bar{\zeta}}}$ to estimate the posterior distribution given the original critical tasks $P^*(\cdot|\{\mathcal{T}_i^{\text{cri}}\}_{i=1}^{N^{\text{cri}}}) = \frac{1}{N^{\bar{\zeta}}}\sum_{j=1}^{N^{\bar{\zeta}}} P^*(\cdot|\{\bar{\mathcal{T}}_i^{\text{cri}}(\lambda_{i,k}^{\bar{\zeta}_j})\}_{i=1}^{N^{\text{cri}}})$. We then solve the upper-level problem in (6) via gradient ascent (line 4). In the following, we elaborate how we solve the lower-level and upper-level problems in (6).

---

**Algorithm 1** Explainable meta reinforcement learning to improve generalization (XMRL-G)

---

**Input**: Initial mixture coefficient distribution $P_{\phi_{\lambda,0}}(\lambda)$ and meta-parameter distribution $P_{\phi_0}(\theta)$, training tasks $\{\mathcal{T}_i^{\text{tr}}\}_{i=1}^{N^{\text{tr}}}$, and poorly-adapted tasks $\{\mathcal{T}_i^{\text{poor}}\}_{i=1}^{N^{\text{poor}}}$.
**Output**: Learned mixture coefficient distribution $P_{\phi_{\lambda,K}}(\lambda)$ and meta-parameter distribution $P_{\phi^*(\{\lambda_{i,K}\}_{i=1}^{N^{\text{cri}}})}(\theta)$.

1: Generate the explanation (i.e., the critical tasks $\{\mathcal{T}_i^{\text{cri}}\}_{i=1}^{N^{\text{cri}}}$) using the algorithm in Appendix A.
2: **for** $k = 0, \cdots, K-1$ **do**
3:     Sample $N^{\bar{\zeta}}$ sets of $\{\lambda_{i,k}^{\bar{\zeta}_j}\}_{i=1}^{N^{\text{cri}}}$ and compute the distribution parameter $\phi^*(\{\lambda_{i,k}^{\bar{\zeta}_j}\}_{i=1}^{N^{\text{cri}}})$ such that
$P^*(\theta|\{\bar{\mathcal{T}}_i^{\text{cri}}(\lambda_{i,k}^{\bar{\zeta}_j})\}_{i=1}^{N^{\text{cri}}}) = P_{\phi^*(\{\lambda_{i,k}^{\bar{\zeta}_j}\}_{i=1}^{N^{\text{cri}}})}(\theta)$ for each set $\{\lambda_{i,k}^{\bar{\zeta}_j}\}_{i=1}^{N^{\text{cri}}}$. Estimate $P^*(\cdot|\{\mathcal{T}_i^{\text{cri}}\}_{i=1}^{N^{\text{cri}}})$
$= \frac{1}{N^{\bar{\zeta}}}\sum_{j=1}^{N^{\bar{\zeta}}} P^*(\cdot|\{\bar{\mathcal{T}}_i^{\text{cri}}(\lambda_{i,k}^{\bar{\zeta}_j})\}_{i=1}^{N^{\text{cri}}})$.
4:     Compute the hyper-gradient $g_{\phi_{\lambda,k}}$ in Lemma 1 and update the mixture coefficient distribution parameter $\phi_{\lambda,k+1} = \phi_{\lambda,k} + \beta g_{\phi_{\lambda,k}}$.
5: **end for**

---

**Solve the lower-level problem** (line 3). To solve the lower-level problem (5), we use a Gaussian distribution to parameterize $P_\phi(\theta)$ and thus the distribution parameter $\phi = (\mu, \Sigma)$ includes a mean vector $\mu$ and a covariance matrix $\Sigma = \sigma\sigma^\top$. We can reparameterize $\theta$ via $\theta = \mu + \sigma \circ \zeta$ where $\zeta \sim \mathcal{N}(0, I)$ draws from a standard Gaussian distribution and $\circ$ is component-wise multiplication. Given $\{\lambda_i^{\bar{\zeta}_j}\}_{i=1}^{N^{\text{cri}}}$, the gradient of problem (5) is $\nabla_\phi E_{p_\phi(\theta)}\left[L(\theta, \{\bar{\mathcal{T}}_i^{\text{cri}}(\lambda_i^{\bar{\zeta}_j})\}_{i=1}^{N^{\text{cri}}}, \{\mathcal{T}_i^{\text{tr}}\}_{i=1}^{N^{\text{tr}}-N^{\text{cri}}})\right] = E_{\zeta \sim \mathcal{N}(0,I)}\left[\nabla_\phi\theta \cdot \nabla_\theta L(\theta, \{\bar{\mathcal{T}}_i^{\text{cri}}(\lambda_i^{\bar{\zeta}_j})\}_{i=1}^{N^{\text{cri}}}, \{\mathcal{T}_i^{\text{tr}}\}_{i=1}^{N^{\text{tr}}-N^{\text{cri}}})\right]$, and we can use $N^\zeta$ samples $\zeta_j \sim \mathcal{N}(0,I)$ to estimate the gradient:

$$g_\phi = \frac{1}{N^\zeta}\sum_{j=1}^{N^\zeta} \nabla_\phi\theta_j \cdot \nabla_\theta L(\theta_j, \{\bar{\mathcal{T}}_i^{\text{cri}}(\lambda_i^{\bar{\zeta}_j})\}_{i=1}^{N^{\text{cri}}}, \{\mathcal{T}_i^{\text{tr}}\}_{i=1}^{N^{\text{tr}}-N^{\text{cri}}}), \tag{7}$$

where $\theta_j = \mu + \sigma \circ \zeta_j$, $\nabla_\phi\theta_j$ is the gradient of $\theta_j$ with respect to the Gaussian distribution parameter $(\mu, \sigma)$, and $\nabla_\theta L(\theta_j, \{\bar{\mathcal{T}}_i^{\text{cri}}(\lambda_i^{\bar{\zeta}_j})\}_{i=1}^{N^{\text{cri}}}, \{\mathcal{T}_i^{\text{tr}}\}_{i=1}^{N^{\text{tr}}-N^{\text{cri}}})$ is the meta-gradient. Note that the meta-gradient $\nabla_\theta L(\theta_j, \{\bar{\mathcal{T}}_i^{\text{cri}}(\lambda_i^{\bar{\zeta}_j})\}_{i=1}^{N^{\text{cri}}}, \{\mathcal{T}_i^{\text{tr}}\}_{i=1}^{N^{\text{tr}}-N^{\text{cri}}})$ can be different for different meta-learning

methods because it depends on what the task-specific adaptation $\pi_i^{\mathrm{tr}}(\theta)$ is, i.e., the lower-level problem in (1). We include the expressions of $\nabla_\theta L(\theta_j, \{\bar{\mathcal{T}}_i^{\mathrm{cri}}(\lambda_i^{\bar{\zeta}_j})\}_{i=1}^{N^{\mathrm{cri}}}, \{\mathcal{T}_i^{\mathrm{tr}}\}_{i=1}^{N^{\mathrm{tr}}-N^{\mathrm{cri}}})$ for several major meta-learning methods in Appendix F.1. We use gradient ascent to solve the lower-level problem (5) to get $\phi^*(\{\lambda_i^{\bar{\zeta}_j}\}_{i=1}^{N^{\mathrm{cri}}}) = (\mu^*(\{\lambda_i^{\bar{\zeta}_j}\}_{i=1}^{N^{\mathrm{cri}}}), \sigma^*(\{\lambda_i^{\bar{\zeta}_j}\}_{i=1}^{N^{\mathrm{cri}}}))$, which is the learned distribution parameter such that $P^*(\theta|\{\bar{\mathcal{T}}_i^{\mathrm{cri}}(\lambda_i^{\bar{\zeta}_j})\}_{i=1}^{N^{\mathrm{cri}}}) = P_{\phi^*(\{\lambda_i^{\bar{\zeta}_j}\}_{i=1}^{N^{\mathrm{cri}}})}(\theta)$. We compute $N^{\bar{\zeta}}$ posterior distributions $\{P^*(\cdot|\{\bar{\mathcal{T}}_i^{\mathrm{cri}}(\lambda_{i,k}^{\bar{\zeta}_j})\}_{i=1}^{N^{\mathrm{cri}}})\}_{\bar{\zeta}_j=1}^{N^{\bar{\zeta}}}$ for $N^{\bar{\zeta}}$ sets of mixture coefficients $\{\{\lambda_{i,k}^{\bar{\zeta}_i}\}_{i=1}^{N^{\mathrm{cri}}}\}_{\bar{\zeta}_j=1}^{N^{\bar{\zeta}}}$, and estimate $P^*(\theta|\{\mathcal{T}_i^{\mathrm{cri}}\}_{i=1}^{N^{\mathrm{cri}}}) = \frac{1}{N^{\bar{\zeta}}}\sum_{j=1}^{N^{\bar{\zeta}}} P^*(\theta|\{\bar{\mathcal{T}}_i^{\mathrm{cri}}(\lambda_i^{\bar{\zeta}_j})\}_{i=1}^{N^{\mathrm{cri}}})$.

**Solve the upper-level problem** (line 4). We use a Gaussian distribution to parameterize $P_{\phi_\lambda}(\lambda)$ where the distribution parameter $\phi_\lambda = (\mu_\lambda, \sigma_\lambda)$ includes a mean $\mu_\lambda$ and a standard deviation $\sigma_\lambda$. Therefore, we can reparameterize each sample $\lambda_i^{\bar{\zeta}_j}$ from $P_{\phi_\lambda}(\lambda)$ via $\lambda_i^{\bar{\zeta}_j} = \mu_\lambda + \sigma_\lambda \bar{\zeta}_{i,j}$ where $\bar{\zeta}_{i,j} \sim \mathcal{N}(0,1)$. To solve the upper-level problem in problem (6), we need to compute the hyper-gradient, i.e., the gradient of the conditional mutual information (4) w.r.t. $\phi_\lambda$.

**Lemma 1.** *Suppose we reparameterize $\lambda_i^{\bar{\zeta}_j}$ via $\lambda_i^{\bar{\zeta}_j} = \mu_\lambda + \sigma_\lambda \bar{\zeta}_{i,j}$, the hyper-gradient can be estimated by* $g_{\phi_\lambda} = \frac{\sum_{j=1}^{N^{\bar{\zeta}}} \nabla_{\phi_\lambda}\sigma^*(\{\lambda_i^{\bar{\zeta}_j}\}_{i=1}^{N^{\mathrm{cri}}})}{|| \sum_{j=1}^{N^{\bar{\zeta}}} \sigma^*(\{\lambda_i^{\bar{\zeta}_j}\}_{i=1}^{N^{\mathrm{cri}}})||} - \frac{1}{N^{\bar{\zeta}}}\sum_{j=1}^{N^{\bar{\zeta}}} \frac{\nabla_{\phi_\lambda}\sigma^*(\{\lambda_i^{\bar{\zeta}_j}\}_{i=1}^{N^{\mathrm{cri}}})}{||\sigma^*(\{\lambda_i^{\bar{\zeta}_j}\}_{i=1}^{N^{\mathrm{cri}}})||}$ *where* $\nabla_{\phi_\lambda}\sigma^*(\{\lambda_i^{\bar{\zeta}_j}\}_{i=1}^{N^{\mathrm{cri}}}) =$
$-\left[\nabla_{\sigma\sigma}^2 E_{P_{\phi^*}(\theta)}[L(\theta, \{\bar{\mathcal{T}}_i^{\mathrm{cri}}(\lambda_i^{\bar{\zeta}_j})\}_{i=1}^{N^{\mathrm{cri}}}, \{\mathcal{T}_i^{\mathrm{tr}}\}_{i=1}^{N^{\mathrm{tr}}-N^{\mathrm{cri}}})]\right]^{-1} \cdot \nabla_{\sigma\phi_\lambda}^2 E_{P_{\phi^*}(\theta)}[L(\theta, \{\bar{\mathcal{T}}_i^{\mathrm{cri}}(\lambda_i^{\bar{\zeta}_j})\}_{i=1}^{N^{\mathrm{cri}}},$
$\{\mathcal{T}_i^{\mathrm{tr}}\}_{i=1}^{N^{\mathrm{tr}}-N^{\mathrm{cri}}})].$

We include the expression of all the gradients in Appendix F. We solve the upper-level problem in (6) via gradient ascent $\phi_{\lambda,k+1} = \phi_{\lambda,k} + \beta g_{\phi_{\lambda,k}}$ where $\beta$ is the step size.

## 5.3 THEORETICAL ANALYSIS

This part shows that (1) Algorithm 1 converges at the rate of $O(1/\sqrt{K})$, (2) the learned augmentation makes the meta-parameter focus more on the critical tasks, and (3) the generalization over the whole task distribution improves after the augmentation. We start with the following assumption:

**Assumption 1.** *The parameterized meta-policy $\pi_\theta$ satisfies the following:* $||\nabla_\theta \log \pi_\theta(a|s)|| \le C_\theta$ *and* $||\nabla_{\theta\theta}^2 \log \pi_\theta(a|s)|| \le \bar{C}_\theta$ *for any $(s,a) \in \mathcal{S} \times \mathcal{A}$ where $C_\theta$ and $\bar{C}_\theta$ are positive constants.*

Assumption 1 assumes that the parameterized log-policy $\log \pi_\theta$ is $C_\theta$-Lipschitz continuous and $\bar{C}_\theta$-smooth w.r.t. the parameter $\theta$, which is a standard assumption in RL (Kumar et al., 2023; Zhang et al., 2020; Agarwal et al., 2021).

**Theorem 1.** *Suppose Assumption 1 holds and $\beta = \frac{2}{\bar{C}_I \sqrt{K}}$ where $\bar{C}_I$ is a positive constant whose derivation is in Appendix G, then Algorithm 1 converges:* $\frac{1}{K}\sum_{k=0}^{K-1} ||\nabla_{\phi_\lambda} I(\theta; \{\bar{\mathcal{T}}_i^{cri}(\lambda_i \sim P_{\phi_\lambda,k}(\lambda))\}_{i=1}^{N^{cri}}|\{\mathcal{T}_i^{cri}\}_{i=1}^{N^{cri}})||^2 \le O(1/\sqrt{K}).$

Theorem 1 shows that Algorithm 1 converges at the rate of $O(1/\sqrt{K})$. We next show that the learned augmentation makes the meta-parameter focus more on the critical tasks:

**Theorem 2.** *Suppose Assumption 1 holds and $\beta < \frac{2}{C_I}$, then the output $P_{\phi_{\lambda,K}}(\lambda)$ of Algorithm 1 satisfies* $I(\theta; \{\bar{\mathcal{T}}_i^{cri}(\lambda_i \sim P_{\phi_\lambda,K}(\lambda))\}_{i=1}^{N^{cri}}|\{\mathcal{T}_i^{cri}\}_{i=1}^{N^{cri}}) > 0.$

Theorem 2 shows that the augmented critical tasks store additional information in the meta-parameter, and thus the meta-parameter pays more attention to the critical tasks. We next quantify the generalization improvement of the learned augmentation $P_{\phi_{\lambda,K}}(\lambda)$. In specific, we first show that the learned augmentation imposes a quadratic regularization on the meta-parameter $\theta$ in Lemma 2 and then show that the generalization over the task distribution $P(\mathcal{T})$ improves.

To reason about the generalization, we consider the following softmax parameterized meta-policy $\pi_\theta(a|s) = \frac{e^{\theta^\top f(s,a)}}{\sum_{a' \in \mathcal{A}} e^{\theta^\top f(s,a')}}$ where $f(s,a)$ is a feature vector. This policy parameterization is widely adopted in RL (Sutton et al., 1999; Kakade, 2001; Peters & Schaal, 2008). We consider MAML

(Finn et al., 2017; Fallah et al., 2021) as the algorithm to compute the task-specific adaptation $\pi_i^{\mathrm{tr}}(\theta)$, and the task-specific adaptation is also softmax parameterized.

**Lemma 2.** *The second-order approximation of the meta-objective (3) after the task augmentation is* $E_{\lambda_i \sim P_{\phi_{\lambda,K}}}[L(\theta, \{\bar{\mathcal{T}}_i^{\mathrm{cri}}(\lambda_i)\}_{i=1}^{N^{\mathrm{cri}}}, \{\mathcal{T}_i^{\mathrm{tr}}\}_{i=1}^{N^{\mathrm{tr}}-N^{\mathrm{cri}}})] \approx L(\theta, \{\mathcal{T}_i^{\mathrm{tr}}\}_{i=1}^{N^{\mathrm{tr}}}) - \theta^{\top}(\frac{1}{N^{\mathrm{cri}}} \sum_{i=1}^{N^{\mathrm{cri}}} \bar{H}_i^{\mathrm{cri}})\theta$ *where $\bar{H}_i^{\mathrm{cri}}$ is a positive definite matrix whose expression is in Appendix I.*

Lemma 2 shows that the augmented meta-objective (3) imposes a quadratic regularization on the original meta-objective (1). Note that we aim to maximize the meta-objective, therefore this negative quadratic regularization reduces the solution space and thus can lead to a better generalization.

To study the generalization property of this regularization, following (Zhang & Deng, 2021; Yao et al., 2021), we consider the following softmax policy class that is closely related to the dual problem of the regularization: $\mathcal{F}_{\bar{\gamma}} = \{\pi_\theta : \theta^{\top}(E_{i \sim P(\mathcal{T})}[\bar{H}_i])\theta \leq \bar{\gamma}\}$. To quantify the improvement of generalization, we denote the generalization gap by $\mathcal{G}(\mathcal{F}_{\bar{\gamma}}) \triangleq L(\theta, \{\mathcal{T}_i^{\mathrm{tr}}\}_{i=1}^{N^{\mathrm{tr}}}) - E_{i \sim P(\mathcal{T})}[L(\theta, \mathcal{T}_i)]$. The following theorem shows the improvement of generalization:

**Theorem 3.** *Suppose the policy is softmax parameterized (i.e., $\pi_\theta(a|s) = \frac{e^{\theta^{\top} f(s,a)}}{\sum_{a' \in \mathcal{A}} e^{\theta^{\top} f(s,a')}}$) where the feature vector $f(s,a)$ is twice-differentiable and bounded for any $(s,a) \in \mathcal{S} \times \mathcal{A}$, then with probability at least $1 - \delta$, the generalization gap satisfies $|\mathcal{G}(\mathcal{F}_{\bar{\gamma}})| \leq O(\sqrt{\frac{\bar{\gamma}}{N^{\mathrm{tr}}}} + \sqrt{\frac{\log(1/\delta)}{N^{\mathrm{tr}}}})$.*

According to Lemma 2, the quadratic regularization (i.e., $\theta^{\top}(\frac{1}{N^{\mathrm{cri}}} \sum_{i=1}^{N^{\mathrm{cri}}} \bar{H}_i^{\mathrm{cri}})\theta$) imposed by the learned task augmentation encourages a smaller $\bar{\gamma}$. Therefore, according to Theorem 3, the learned task augmentation will lead to a smaller generalization gap and thus improve generalization.

## 6 EXPERIMENT

This section uses two real-world experiments and three MuJoCo experiments to show the effectiveness of Algorithm 1 (XMRL-G), where the first real-world experiment is conducted on a physical drone and the second real-world experiment uses real-world stock market data. We introduce three baselines for comparisons: (1) **Task weighting** (Nguyen et al., 2023): This method learns how to weight different training tasks in order to improve generalization. (2) **Meta augmentation** (Yao et al., 2021): This method uses a pre-defined distribution of $\lambda$ to mix the data of each training task to improve generalization. (3) **Meta regularization** (Wang et al., 2023): This method adds quadratic regularization to the upper level and inverted regularization to the lower level to improve generalization. We use MAML (Finn et al., 2017; Fallah et al., 2021) as the baseline meta-RL algorithm.

### 6.1 DRONE NAVIGATION WITH OBSTACLES

We conduct a navigation experiment (Figure 1) on an AR.Drone 2.0 where the drone (in the yellow bounding box) wants to navigate to the goal (in the green bounding box) while avoiding the obstacle (in the red bounding box). We use an indoor motion capture system "Vicon" to record the location of the drone and send this location information to the drone. For different navigation tasks, we change the locations of the goal and the obstacle. The reward function is designed to be positive at the goal, negative at the obstacle, and zero otherwise. We use success rate (i.e., the rate of successfully reaching the goal and avoiding collision with the obstacle) as the metric to evaluate the RL performance. We

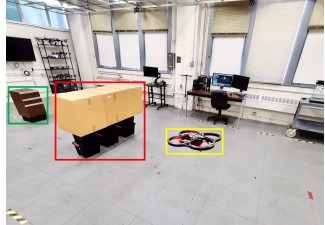

Figure 1: Drone navigation

use 50 training tasks to train a meta-policy and find 5 poorly-adapted tasks. To evaluate the generalization, we randomly generate 20 test tasks and record the mean and standard deviation of success rate in the second row in Table 1. The experiment details are in Appendix K.1.

### 6.2 STOCK MARKET

RL to train a stock trading agent has been widely studied in AI for finance (Deng et al., 2016; Liu & Zhu, 2024b). We use the real-world data of 30 constituent stocks in Dow Jones Industrial Average

Table 1: Experiment results.

|  | MAML | XMRL-G | Task weighting | Meta augmentation | Meta regularization |
|---|---|---|---|---|---|
| Drone | $0.87 \pm 0.01$ | $0.96 \pm 0.02$ | $0.88 \pm 0.02$ | $0.91 \pm 0.02$ | $0.91 \pm 0.02$ |
| Stock Market | $359.13 \pm 18.63$ | $426.36 \pm 17.15$ | $371.88 \pm 17.25$ | $389.17 \pm 12.66$ | $362.53 \pm 14.27$ |
| HalfCheetah | $-68.89 \pm 4.36$ | $-53.88 \pm 5.21$ | $-66.77 \pm 6.38$ | $-63.49 \pm 4.07$ | $-61.15 \pm 3.82$ |
| Hopper | $-23.24 \pm 5.71$ | $-12.50 \pm 2.37$ | $-19.35 \pm 4.12$ | $-22.37 \pm 4.65$ | $-16.23 \pm 2.03$ |
| Walker | $-82.18 \pm 6.64$ | $-55.76 \pm 5.01$ | $-76.86 \pm 5.29$ | $-67.51 \pm 4.83$ | $-73.25 \pm 4.27$ |

from 2021-01-01 to 2022-01-01. We use a benchmark "FinRL" (Liu et al., 2021) to configure the real-world stock data into an MDP environment. The RL agent trades stocks on every stock market opening day in order to maximize profit as well as avoid taking risks. The reward function is defined as $p_1 - p_2$ where $p_1$ is the profit which is the money earned from trading stocks subtracting the transaction cost, and $p_2$ models the preference of whether willing to take risks. In specific, $p_2$ is positive if the investor buys stocks whose turbulence indices are larger than a certain turbulence threshold, and zero otherwise. The value of $p_2$ depends on the type and amount of the trading stocks. The turbulence index measures the risk of buying a stock (Liu et al., 2021), and a lower turbulence threshold means that the RL agent is less willing to take risks. The turbulence thresholds for different RL tasks are different. We use 50 training tasks to learn a meta-policy and find 5 poorly-adapted tasks. We use 20 test tasks to evaluate the generalization. We include the details in Appendix K.2 and the results of cumulative reward in the third row in Table 1.

### 6.3 MuJoCo

We consider the target velocity problem (Finn et al., 2017; Rakelly et al., 2019; Lin et al., 2020b; Liu & Zhu, 2023) for three MuJoCo robots: HalfCheetah, Walker, and Hopper. In specific, the robots aim to maintain a target velocity in each task and the target velocity of different tasks is different. The reward function is designed as $-|v - v_{\text{target}}|$ (as in Finn et al. (2017)) where $v$ is the current robot velocity and $v_{\text{target}}$ is the target velocity. We use 50 training tasks to learn a meta-policy and find 5 poorly-adapted tasks. We use 20 test tasks to evaluate the generalization. We include the details in Appendix K.3 and the results of cumulative reward in the fourth to sixth rows in Table 1.

Table 1 shows that our proposed method can significantly improve the generalization of MAML and outperform the other three baselines.

**Evaluation of the explanation**. We also aim to evaluate the fidelity and usefulness of our explanation. Fidelity is a widely-used metric in explainable RL (Guo et al., 2021b; Cheng et al., 2024) to evaluate the correctness of the explanation. The fidelity in our setting means whether the identified critical tasks $\{\mathcal{T}_i^{\text{cri}}\}_{i=1}^{N^{\text{cri}}}$ are indeed the most important training tasks to achieve high cumulative reward on the poorly-adapted tasks $\{\mathcal{T}_i^{\text{poor}}\}_{i=1}^{N^{\text{poor}}}$. To evaluate fidelity, we train a meta-policy over the critical tasks and compare its performance on the poorly-adapted tasks with a meta-policy trained on $N^{\text{cri}}$ randomly-sampled training tasks. The usefulness means whether our explanation can help improve generalization. To evaluate the usefulness, we randomly pick $N^{\text{cri}}$ training tasks and use our augmentation method to augment these $N^{\text{cri}}$ training tasks to train a meta-policy. We compare the generalization performance of this meta-policy with XMRL-G. We include the results in Appendix K.4, and the results show that our explanation has high fidelity and usefulness.

## 7 CONCLUSION

This paper proposes the first method that uses explainable meta-RL to improve generalization of meta-RL. The proposed method has two parts where the first part explains why the learned meta-policy does not adapt well to certain tasks by identifying the critical training tasks that the meta-policy does not pay enough attention to, and the second part formulates a bi-level optimization problem to learn how to augment the critical tasks such that the meta-policy can best pay attention to the critical tasks. We theoretically guarantee that the learned augmentation can improve generalization over the whole task distribution. Two real-world experiments and three MuJoCo experiments are used to show that our method outperforms state-of-the-art baselines.

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

## A ALGORITHM TO FIND THE CRITICAL TASKS

Recall from Section 4 that we aim to learn a weight vector $\omega$ by solving the problem (2) where each component $\omega_i$ of the weight vector captures the importance of the corresponding training task $\mathcal{T}_i^{\text{tr}}$. The higher the weight value $\omega_i$ is, the more important the corresponding training task $\mathcal{T}_i^{\text{tr}}$ is. Therefore, the top $N^{\text{cri}}$ training tasks with highest weight values are the $N^{\text{cri}}$ critical tasks we aim to identify. The problem (2) is as follows:

$$\max_{\omega} \ L(\theta^*(\omega), \{\mathcal{T}_i^{\text{poor}}\}_{i=1}^{N^{\text{poor}}}) \quad \text{s.t.} \ \theta^*(\omega) = \arg\max_{\theta} \sum_{i=1}^{N^{\text{tr}}} \omega_i J_i^{\text{tr}}(\pi_i^{\text{tr}}(\theta)).$$

We use Algorithm 2 to solve this problem where at each iteration $\bar{k}$, we first solve the lower-level problem in (2) to get $\theta^*(\omega)$ and then solve the upper-level problem (2) via gradient ascent.

---

**Algorithm 2** Identifying the critical tasks

---

**Input**: Training tasks $\{\mathcal{T}_i^{\text{tr}}\}_{i=1}^{N^{\text{tr}}}$, poorly-adapted tasks $\{\mathcal{T}_i^{\text{poor}}\}_{i=1}^{N^{\text{poor}}}$, and initial weight vector $\omega$.
**Output**: Learned weight vector $\omega_{\bar{K}}$.
 1: **for** $\bar{k} = 0, \cdots, \bar{K} - 1$ **do**
 2:     Solve the lower-level problem via gradient ascent to get $\theta^*(\omega)$.
 3:     Compute the hyper-gradient $g_{\omega_{\bar{k}}}$ in Lemma 3 and update the weight $\omega_{\bar{k}+1} = \omega_{\bar{k}} + \alpha_{\bar{k}} g_{\omega_{\bar{k}}}$.
 4: **end for**

---

**Solve the lower-level problem**. We use gradient ascent to solve the lower-level problem where the gradient is $\sum_{i=1}^{N^{\text{tr}}} \omega_i \nabla_\theta J_i^{\text{tr}}(\pi_i^{\text{tr}}(\theta))$ and the expression of $\nabla_\theta J_i^{\text{tr}}(\pi_i^{\text{tr}}(\theta))$ can be found in Appendix F.1.

**Solve the upper-level problem**. To solve the upper-level problem, we need to compute the hyper-gradient $g_\omega$.

**Lemma 3.** *The hyper-gradient is:*

$g_\omega =$

$$- \left[ \nabla_\omega \sum_{i=1}^{N^{\text{tr}}} \omega_i \nabla_\theta J_i^{\text{tr}}(\pi_i^{\text{tr}}(\theta^*(\omega))) \right] \left[ \sum_{i=1}^{N^{\text{tr}}} \omega_i \nabla_{\theta\theta}^2 J_i^{\text{tr}}(\pi_i^{\text{tr}}(\theta^*(\omega))) \right]^{-1} \left[ \sum_{i=1}^{N^{\text{poor}}} \nabla_\theta J_i^{\text{poor}}(\pi_i^{\text{poor}}(\theta^*(\omega))) \right],$$

*where the derivation is in Appendix A.1.*

We use $\bar{K}$-step gradient ascent $\omega_{\bar{k}+1} = \omega_{\bar{k}} + \alpha_{\bar{k}} g_{\omega_{\bar{k}}}$ to solve the problem (2) to get the learned weight $\omega_{\bar{K}}$. Each component $\omega_{\bar{K},i}$ captures the importance of the corresponding training task $\mathcal{T}_i^{\text{tr}}$. We pick the top $N^{\text{cri}}$ training tasks with the highest weight value as the critical tasks.

### A.1 PROOF OF LEMMA 3

Since $\theta^*(\omega) = \arg\max_{\theta} \sum_{i=1}^{N^{\text{tr}}} \omega_i J_i^{\text{tr}}(\pi_i^{\text{tr}}(\theta))$, then $\nabla_\theta \sum_{i=1}^{N^{\text{tr}}} \omega_i J_i^{\text{tr}}(\pi_i^{\text{tr}}(\theta^*(\omega))) = 0$. Take gradient w.r.t. $\omega$ on both sides, we have that

$$\nabla_{\omega\theta}^2 \sum_{i=1}^{N^{\text{tr}}} \omega_i J_i^{\text{tr}}(\pi_i^{\text{tr}}(\theta^*(\omega))) + \left( \nabla_\omega \theta^*(\omega) \right)^\top \left[ \nabla_{\theta\theta}^2 \sum_{i=1}^{N^{\text{tr}}} \omega_i J_i^{\text{tr}}(\pi_i^{\text{tr}}(\theta^*(\omega))) \right] = 0,$$

$$\Rightarrow \nabla_\omega \theta^*(\omega) = \left[ \nabla^2_{\theta\theta} \sum_{i=1}^{N^{\mathrm{tr}}} \omega_i J_i^{\mathrm{tr}}(\pi_i^{\mathrm{tr}}(\theta^*(\omega))) \right]^{-1} \left[ \nabla^2_{\theta\omega} \sum_{i=1}^{N^{\mathrm{tr}}} \omega_i J_i^{\mathrm{tr}}(\pi_i^{\mathrm{tr}}(\theta^*(\omega))) \right]. \tag{8}$$

Therefore, we have that

$$\nabla_\omega L(\theta^*(\omega), \{\mathcal{T}_i^{\mathrm{poor}}\}_{i=1}^{N^{\mathrm{poor}}}) = \left( \nabla_\omega \theta^*(\omega) \right)^\top \nabla_\theta L(\theta^*(\omega), \{\mathcal{T}_i^{\mathrm{poor}}\}_{i=1}^{N^{\mathrm{poor}}}),$$

$$\overset{(a)}{=} \left[ \nabla^2_{\omega\theta} \sum_{i=1}^{N^{\mathrm{tr}}} \omega_i J_i^{\mathrm{tr}}(\pi_i^{\mathrm{tr}}(\theta^*(\omega))) \right] \left[ \nabla^2_{\theta\theta} \sum_{i=1}^{N^{\mathrm{tr}}} \omega_i J_i^{\mathrm{tr}}(\pi_i^{\mathrm{tr}}(\theta^*(\omega))) \right]^{-1} \nabla_\theta L(\theta^*(\omega), \{\mathcal{T}_i^{\mathrm{poor}}\}_{i=1}^{N^{\mathrm{poor}}}),$$

where $(a)$ follows (8).

## B  THE TASK AUGMENTATION DOES NOT COMPROMISE THE PERFORMANCE ON THE NON-CRITICAL TASKS

This section shows that the task augmentation does not compromise the performance on the non-critical tasks. In brief, we prove that the mutual information between the meta-parameter and the non-critical tasks remains unchanged even if the mutual information between the meta-parameter and the critical tasks increases after task augmentation. Since the task information of the non-critical tasks stored in the meta-parameter does not change after augmentation, the performance on the non-critical tasks is not compromised.

Suppose we augment the critical tasks $\{\mathcal{T}_i^{\mathrm{cri}}\}_{i=1}^{N^{\mathrm{cri}}}$ to $\{\bar{\mathcal{T}}_i^{\mathrm{cri}}\}_{i=1}^{N^{\mathrm{cri}}}$. Note that the difference between $\{\mathcal{T}_i^{\mathrm{cri}}\}_{i=1}^{N^{\mathrm{cri}}}$ and $\{\bar{\mathcal{T}}_i^{\mathrm{cri}}\}_{i=1}^{N^{\mathrm{cri}}}$ is that they have different distributions, i.e., $P(\{\mathcal{T}_i^{\mathrm{cri}}\}_{i=1}^{N^{\mathrm{cri}}})$ and $P(\{\bar{\mathcal{T}}_i^{\mathrm{cri}}\}_{i=1}^{N^{\mathrm{cri}}})$. Therefore, we use $A$ to generally represent the critical tasks (either before augmentation or after augmentation), and use $P(A = \{\mathcal{T}_i^{\mathrm{cri}}\}_{i=1}^{N^{\mathrm{cri}}})$ and $P(A = \{\bar{\mathcal{T}}_i^{\mathrm{cri}}\}_{i=1}^{N^{\mathrm{cri}}})$ to respectively denote that $A$ follows the distribution of $\{\mathcal{T}_i^{\mathrm{cri}}\}_{i=1}^{N^{\mathrm{cri}}}$ and $A$ follows the distribution of $\{\bar{\mathcal{T}}_i^{\mathrm{cri}}\}_{i=1}^{N^{\mathrm{cri}}}$. We now quantify the change of the mutual information between the meta-parameter and the non-critical tasks $\{\mathcal{T}_i^{\mathrm{tr}}\}_{i=1}^{N^{\mathrm{tr}}-N^{\mathrm{cri}}}$:

$$I(\theta; \{\mathcal{T}_i^{\mathrm{tr}}\}_{i=1}^{N^{\mathrm{tr}}-N^{\mathrm{cri}}} | \{\bar{\mathcal{T}}_i^{\mathrm{cri}}\}_{i=1}^{N^{\mathrm{cri}}}) - I(\theta; \{\mathcal{T}_i^{\mathrm{tr}}\}_{i=1}^{N^{\mathrm{tr}}-N^{\mathrm{cri}}} | \{\mathcal{T}_i^{\mathrm{cri}}\}_{i=1}^{N^{\mathrm{cri}}}),$$

$$\overset{(a)}{=} \int P(\theta, \{\mathcal{T}_i^{\mathrm{tr}}\}_{i=1}^{N^{\mathrm{tr}}-N^{\mathrm{cri}}}, \{\bar{\mathcal{T}}_i^{\mathrm{cri}}\}_{i=1}^{N^{\mathrm{cri}}}) \cdot$$

$$\log \frac{P(\theta, \{\mathcal{T}_i^{\mathrm{tr}}\}_{i=1}^{N^{\mathrm{tr}}-N^{\mathrm{cri}}} | \{\bar{\mathcal{T}}_i^{\mathrm{cri}}\}_{i=1}^{N^{\mathrm{cri}}})}{P(\theta | \{\bar{\mathcal{T}}_i^{\mathrm{cri}}\}_{i=1}^{N^{\mathrm{cri}}}) P(\{\mathcal{T}_i^{\mathrm{tr}}\}_{i=1}^{N^{\mathrm{tr}}-N^{\mathrm{cri}}} | \{\bar{\mathcal{T}}_i^{\mathrm{cri}}\}_{i=1}^{N^{\mathrm{cri}}})} d\theta (d\{\mathcal{T}_i^{\mathrm{tr}}\}_{i=1}^{N^{\mathrm{tr}}-N^{\mathrm{cri}}}) (d\{\bar{\mathcal{T}}_i^{\mathrm{cri}}\}_{i=1}^{N^{\mathrm{cri}}})$$

$$- \int P(\theta, \{\mathcal{T}_i^{\mathrm{tr}}\}_{i=1}^{N^{\mathrm{tr}}-N^{\mathrm{cri}}}, \{\mathcal{T}_i^{\mathrm{cri}}\}_{i=1}^{N^{\mathrm{cri}}}) \cdot$$

$$\log \frac{P(\theta, \{\mathcal{T}_i^{\mathrm{tr}}\}_{i=1}^{N^{\mathrm{tr}}-N^{\mathrm{cri}}} | \{\mathcal{T}_i^{\mathrm{cri}}\}_{i=1}^{N^{\mathrm{cri}}})}{P(\theta | \{\mathcal{T}_i^{\mathrm{cri}}\}_{i=1}^{N^{\mathrm{cri}}}) P(\{\mathcal{T}_i^{\mathrm{tr}}\}_{i=1}^{N^{\mathrm{tr}}-N^{\mathrm{cri}}} | \{\mathcal{T}_i^{\mathrm{cri}}\}_{i=1}^{N^{\mathrm{cri}}})} d\theta (d\{\mathcal{T}_i^{\mathrm{tr}}\}_{i=1}^{N^{\mathrm{tr}}-N^{\mathrm{cri}}}) (d\{\mathcal{T}_i^{\mathrm{cri}}\}_{i=1}^{N^{\mathrm{cri}}})$$

$$= \int P(\theta, \{\mathcal{T}_i^{\mathrm{tr}}\}_{i=1}^{N^{\mathrm{tr}}-N^{\mathrm{cri}}}, \{\bar{\mathcal{T}}_i^{\mathrm{cri}}\}_{i=1}^{N^{\mathrm{cri}}}) \log \frac{P(\theta | \{\mathcal{T}_i^{\mathrm{tr}}\}_{i=1}^{N^{\mathrm{tr}}-N^{\mathrm{cri}}}, \{\bar{\mathcal{T}}_i^{\mathrm{cri}}\}_{i=1}^{N^{\mathrm{cri}}})}{P(\theta | \{\bar{\mathcal{T}}_i^{\mathrm{cri}}\}_{i=1}^{N^{\mathrm{cri}}})} d\theta (d\{\mathcal{T}_i^{\mathrm{tr}}\}_{i=1}^{N^{\mathrm{tr}}-N^{\mathrm{cri}}}) (d\{\bar{\mathcal{T}}_i^{\mathrm{cri}}\}_{i=1}^{N^{\mathrm{cri}}})$$

$$- \int P(\theta, \{\mathcal{T}_i^{\mathrm{tr}}\}_{i=1}^{N^{\mathrm{tr}}-N^{\mathrm{cri}}}, \{\mathcal{T}_i^{\mathrm{cri}}\}_{i=1}^{N^{\mathrm{cri}}}) \log \frac{P(\theta | \{\mathcal{T}_i^{\mathrm{tr}}\}_{i=1}^{N^{\mathrm{tr}}-N^{\mathrm{cri}}}, \{\mathcal{T}_i^{\mathrm{cri}}\}_{i=1}^{N^{\mathrm{cri}}})}{P(\theta | \{\mathcal{T}_i^{\mathrm{cri}}\}_{i=1}^{N^{\mathrm{cri}}})} d\theta (d\{\mathcal{T}_i^{\mathrm{tr}}\}_{i=1}^{N^{\mathrm{tr}}-N^{\mathrm{cri}}}) (d\{\mathcal{T}_i^{\mathrm{cri}}\}_{i=1}^{N^{\mathrm{cri}}}),$$

$$\overset{(b)}{=} \int P(\theta | \{\mathcal{T}_i^{\mathrm{tr}}\}_{i=1}^{N^{\mathrm{tr}}-N^{\mathrm{cri}}}, A) P(\{\mathcal{T}_i^{\mathrm{tr}}\}_{i=1}^{N^{\mathrm{tr}}-N^{\mathrm{cri}}}) P(A = \{\bar{\mathcal{T}}_i^{\mathrm{cri}}\}_{i=1}^{N^{\mathrm{cri}}}) \cdot$$

$$\log \frac{P(\theta | \{\mathcal{T}_i^{\mathrm{tr}}\}_{i=1}^{N^{\mathrm{tr}}-N^{\mathrm{cri}}}, A)}{P(\theta | A)} d\theta (d\{\mathcal{T}_i^{\mathrm{tr}}\}_{i=1}^{N^{\mathrm{tr}}-N^{\mathrm{cri}}}) (dA)$$

$$- \int P(\theta | \{\mathcal{T}_i^{\mathrm{tr}}\}_{i=1}^{N^{\mathrm{tr}}-N^{\mathrm{cri}}}, A) P(\{\mathcal{T}_i^{\mathrm{tr}}\}_{i=1}^{N^{\mathrm{tr}}-N^{\mathrm{cri}}}) P(A = \{\mathcal{T}_i^{\mathrm{cri}}\}_{i=1}^{N^{\mathrm{cri}}}) \cdot$$

$$\log \frac{P(\theta | \{\mathcal{T}_i^{\mathrm{tr}}\}_{i=1}^{N^{\mathrm{tr}}-N^{\mathrm{cri}}}, A)}{P(\theta | A)} d\theta (d\{\mathcal{T}_i^{\mathrm{tr}}\}_{i=1}^{N^{\mathrm{tr}}-N^{\mathrm{cri}}}) (dA),$$

$$= \int P(\theta|\{\mathcal{T}_i^{\text{tr}}\}_{i=1}^{N^{\text{tr}}-N^{\text{cri}}}, A) P(\{\mathcal{T}_i^{\text{tr}}\}_{i=1}^{N^{\text{tr}}-N^{\text{cri}}}) \Big[ P(A = \{\bar{\mathcal{T}}_i^{\text{cri}}\}_{i=1}^{N^{\text{cri}}}) - P(A = \{\mathcal{T}_i^{\text{cri}}\}_{i=1}^{N^{\text{cri}}}) \Big] \cdot$$

$$\log \frac{P(\theta|\{\mathcal{T}_i^{\text{tr}}\}_{i=1}^{N^{\text{tr}}-N^{\text{cri}}}, A)}{P(\theta|A)} d\theta (d\{\mathcal{T}_i^{\text{tr}}\}_{i=1}^{N^{\text{tr}}-N^{\text{cri}}})(dA),$$

$$= \int P(\theta|\{\mathcal{T}_i^{\text{tr}}\}_{i=1}^{N^{\text{tr}}-N^{\text{cri}}}, A) P(\{\mathcal{T}_i^{\text{tr}}\}_{i=1}^{N^{\text{tr}}-N^{\text{cri}}}) \Big[ P(A = \{\bar{\mathcal{T}}_i^{\text{cri}}\}_{i=1}^{N^{\text{cri}}}) - P(A = \{\mathcal{T}_i^{\text{cri}}\}_{i=1}^{N^{\text{cri}}}) \Big] \cdot$$

$$\log \frac{P(\theta|\{\mathcal{T}_i^{\text{tr}}\}_{i=1}^{N^{\text{tr}}-N^{\text{cri}}}, A)}{\int P(\theta|\{\mathcal{T}_i^{\text{tr}}\}_{i=1}^{N^{\text{tr}}-N^{\text{cri}}}, A) P(\{\mathcal{T}_i^{\text{tr}}\}_{i=1}^{N^{\text{tr}}-N^{\text{cri}}})(d\{\mathcal{T}_i^{\text{tr}}\}_{i=1}^{N^{\text{tr}}-N^{\text{cri}}})} d\theta (d\{\mathcal{T}_i^{\text{tr}}\}_{i=1}^{N^{\text{tr}}-N^{\text{cri}}})(dA),$$

$$\overset{(c)}{=} \int P(\theta|\{\mathcal{T}_i^{\text{tr}}\}_{i=1}^{N^{\text{tr}}-N^{\text{cri}}}, A) P(\{\mathcal{T}_i^{\text{tr}}\}_{i=1}^{N^{\text{tr}}-N^{\text{cri}}}) \Big[ P(A = \{\bar{\mathcal{T}}_i^{\text{cri}}\}_{i=1}^{N^{\text{cri}}}) - P(A = \{\mathcal{T}_i^{\text{cri}}\}_{i=1}^{N^{\text{cri}}}) \Big] \cdot$$

$$\log 1 d\theta (d\{\mathcal{T}_i^{\text{tr}}\}_{i=1}^{N^{\text{tr}}-N^{\text{cri}}})(dA),$$

$$= 0, \tag{9}$$

where $(a)$ follows the definition of conditional mutual information (Wyner, 1978), $(b)$ follows the fact that the critical tasks and the non-critical tasks are independent (i.e., $P(\theta, \{\mathcal{T}_i^{\text{tr}}\}_{i=1}^{N^{\text{tr}}-N^{\text{cri}}}, A) = P(\theta|\{\mathcal{T}_i^{\text{tr}}\}_{i=1}^{N^{\text{tr}}-N^{\text{cri}}}, A) P(\{\mathcal{T}_i^{\text{tr}}\}_{i=1}^{N^{\text{tr}}-N^{\text{cri}}}, A) = P(\theta|\{\mathcal{T}_i^{\text{tr}}\}_{i=1}^{N^{\text{tr}}-N^{\text{cri}}}, A) P(\{\mathcal{T}_i^{\text{tr}}\}_{i=1}^{N^{\text{tr}}-N^{\text{cri}}}) P(A))$, and $(c)$ follows the fact that the non-critical tasks $\{\mathcal{T}_i^{\text{tr}}\}_{i=1}^{N^{\text{tr}}-N^{\text{cri}}}$ are given and thus $P(\{\mathcal{T}_i^{\text{tr}}\}_{i=1}^{N^{\text{tr}}-N^{\text{cri}}}) = 1$.

From (9), we can see that $I(\theta; \{\mathcal{T}_i^{\text{tr}}\}_{i=1}^{N^{\text{tr}}-N^{\text{cri}}}|\{\bar{\mathcal{T}}_i^{\text{cri}}\}_{i=1}^{N^{\text{cri}}}) - I(\theta; \{\mathcal{T}_i^{\text{tr}}\}_{i=1}^{N^{\text{tr}}-N^{\text{cri}}}|\{\mathcal{T}_i^{\text{cri}}\}_{i=1}^{N^{\text{cri}}}) = 0$, and thus the information of the non-critical tasks stored in the meta-parameter does not change after the task augmentation. Therefore, the performance on the non-critical tasks is not compromised.

## C  EXPRESSION OF THE AUGMENTED STATE-ACTION STATIONARY DISTRIBUTION

The expression of the augmented state-action stationary distribution is $\bar{\rho}^{\pi, \lambda_i}(\bar{s}_{jj'}, \bar{a}_{jj'}) \triangleq \sum_{(s,a),(s',a')\in\mathcal{S}\times\mathcal{A}} \mathbb{1}\{\lambda_i s + (1-\lambda_i)s' = \bar{s}_{jj'}\}[\mathbb{1}\{\lambda_i \geq 0.5\}\rho^\pi(s, \bar{a}_{jj'})\rho^\pi(s', a') + \mathbb{1}\{\lambda_i < 0.5\}\rho^\pi(s, a)\rho^\pi(s', \bar{a}_{jj'})]$. For each $(\bar{s}_{jj'}, \bar{a}_{jj'})$, we sum the joint probability of any two state-action pairs whose mixture combination is $(\bar{s}_{jj'}, \bar{a}_{jj'})$.

## D  DERIVATION OF THE CONDITIONAL MUTUAL INFORMATION

$$I(\theta; \{\bar{\mathcal{T}}_i^{\text{cri}}(\lambda_i \sim P(\lambda))\}_{i=1}^{N^{\text{cri}}}|\{\mathcal{T}_i^{\text{cri}}\}_{i=1}^{N^{\text{cri}}}),$$

$$\overset{(a)}{=} \int P(\theta, \{\bar{\mathcal{T}}_i^{\text{cri}}(\lambda_i \sim P(\lambda))\}_{i=1}^{N^{\text{cri}}}, \{\mathcal{T}_i^{\text{cri}}\}_{i=1}^{N^{\text{cri}}}) \cdot$$

$$\log \frac{P(\theta, \{\bar{\mathcal{T}}_i^{\text{cri}}(\lambda_i \sim P(\lambda))\}_{i=1}^{N^{\text{cri}}}|\{\mathcal{T}_i^{\text{cri}}\}_{i=1}^{N^{\text{cri}}})}{P(\theta|\{\mathcal{T}_i^{\text{cri}}\}_{i=1}^{N^{\text{cri}}}) P(\{\bar{\mathcal{T}}_i^{\text{cri}}(\lambda_i \sim P(\lambda))\}_{i=1}^{N^{\text{cri}}}|\{\mathcal{T}_i^{\text{cri}}\}_{i=1}^{N^{\text{cri}}})} (d\theta)(d\{\bar{\mathcal{T}}_i^{\text{cri}}(\lambda_i \sim P(\lambda))\}_{i=1}^{N^{\text{cri}}})(d\{\mathcal{T}_i^{\text{cri}}\}_{i=1}^{N^{\text{cri}}}),$$

$$= \int P(\theta|\{\bar{\mathcal{T}}_i^{\text{cri}}(\lambda_i)\}_{i=1}^{N^{\text{cri}}}, \{\mathcal{T}_i^{\text{cri}}\}_{i=1}^{N^{\text{cri}}}) P(\lambda) P(\{\mathcal{T}_i^{\text{cri}}\}_{i=1}^{N^{\text{cri}}}) \cdot$$

$$\log \frac{P(\theta, \{\bar{\mathcal{T}}_i^{\text{cri}}(\lambda_i)\}_{i=1}^{N^{\text{cri}}}|\{\mathcal{T}_i^{\text{cri}}\}_{i=1}^{N^{\text{cri}}})}{P(\theta|\{\mathcal{T}_i^{\text{cri}}\}_{i=1}^{N^{\text{cri}}}) P(\{\bar{\mathcal{T}}_i^{\text{cri}}(\lambda_i)\}_{i=1}^{N^{\text{cri}}}|\{\mathcal{T}_i^{\text{cri}}\}_{i=1}^{N^{\text{cri}}})} (d\theta)(d\{\bar{\mathcal{T}}_i^{\text{cri}}(\lambda_i)\}_{i=1}^{N^{\text{cri}}})(d\{\mathcal{T}_i^{\text{cri}}\}_{i=1}^{N^{\text{cri}}}),$$

$$= \int P(\theta|\{\bar{\mathcal{T}}_i^{\text{cri}}(\lambda_i)\}_{i=1}^{N^{\text{cri}}}, \{\mathcal{T}_i^{\text{cri}}\}_{i=1}^{N^{\text{cri}}}) P(\lambda) P(\{\mathcal{T}_i^{\text{cri}}\}_{i=1}^{N^{\text{cri}}}) \cdot$$

$$\log \frac{P(\theta|\{\bar{\mathcal{T}}_i^{\text{cri}}(\lambda_i)\}_{i=1}^{N^{\text{cri}}}, \{\mathcal{T}_i^{\text{cri}}\}_{i=1}^{N^{\text{cri}}})}{P(\theta|\{\mathcal{T}_i^{\text{cri}}\}_{i=1}^{N^{\text{cri}}})} (d\theta)(d\lambda_i)(d\{\mathcal{T}_i^{\text{cri}}\}_{i=1}^{N^{\text{cri}}}),$$

$$\overset{(b)}{=} \int P(\theta|\{\bar{\mathcal{T}}_i^{\text{cri}}(\lambda_i)\}_{i=1}^{N^{\text{cri}}}) P(\lambda) \log \frac{P(\theta|\{\bar{\mathcal{T}}_i^{\text{cri}}(\lambda_i)\}_{i=1}^{N^{\text{cri}}})}{P(\theta|\{\mathcal{T}_i^{\text{cri}}\}_{i=1}^{N^{\text{cri}}})} (d\theta)(d\lambda_i)(d\{\mathcal{T}_i^{\text{cri}}\}_{i=1}^{N^{\text{cri}}}),$$

$$= E_{\lambda_i \in [0,1], \lambda_i \sim P(\lambda), \theta \sim P(\cdot|\{\bar{\mathcal{T}}_i^{\mathrm{cri}}(\lambda_i)\}_{i=1}^{N^{\mathrm{cri}}})} \left[ \log \frac{P(\theta|\{\bar{\mathcal{T}}_i^{\mathrm{cri}}(\lambda_i)\}_{i=1}^{N^{\mathrm{cri}}})}{P(\theta|\{\mathcal{T}_i^{\mathrm{cri}}\}_{i=1}^{N^{\mathrm{cri}}})} \right],$$

where $(a)$ follows the definition of conditional mutual information (Wyner, 1978) and $(b)$ follows the fact that $P(\theta|\{\bar{\mathcal{T}}_i^{\mathrm{cri}}(\lambda_i)\}_{i=1}^{N^{\mathrm{cri}}}, \{\mathcal{T}_i^{\mathrm{cri}}\}_{i=1}^{N^{\mathrm{cri}}}) = P(\theta|\{\bar{\mathcal{T}}_i^{\mathrm{cri}}(\lambda_i)\}_{i=1}^{N^{\mathrm{cri}}})$ because the meta-parameter is trained on the augmented critical tasks $\{\bar{\mathcal{T}}_i^{\mathrm{cri}}(\lambda_i)\}_{i=1}^{N^{\mathrm{cri}}}$.

# E  PROOF OF LEMMA 1

Recall from (4) that

$$I(\theta; \{\bar{\mathcal{T}}_i^{\mathrm{cri}}(\lambda_i \sim P_{\phi_\lambda}(\lambda))\}_{i=1}^{N^{\mathrm{cri}}} | \{\mathcal{T}_i^{\mathrm{cri}}\}_{i=1}^{N^{\mathrm{cri}}}),$$

$$= E_{\lambda_i \in [0,1], \lambda_i \sim P_{\phi_\lambda}(\lambda), \theta \sim P^*(\cdot|\{\bar{\mathcal{T}}_i^{\mathrm{cri}}(\lambda_i)\}_{i=1}^{N^{\mathrm{cri}}})} \left[ \log \frac{P^*(\theta|\{\bar{\mathcal{T}}_i^{\mathrm{cri}}(\lambda_i)\}_{i=1}^{N^{\mathrm{cri}}})}{P^*(\theta|\{\mathcal{T}_i^{\mathrm{cri}}\}_{i=1}^{N^{\mathrm{cri}}})} \right].$$

Since $P_{\phi^*(\{\lambda_i^{\bar{\zeta}_j}\}_{i=1}^{N^{\mathrm{cri}}})}(\theta)$ is Gaussian distribution, we have that

$$I(\theta; \{\bar{\mathcal{T}}_i^{\mathrm{cri}}(\lambda_i \sim P_{\phi_\lambda}(\lambda))\}_{i=1}^{N^{\mathrm{cri}}} | \{\mathcal{T}_i^{\mathrm{cri}}\}_{i=1}^{N^{\mathrm{cri}}}),$$

$$= E_{\lambda_i, \theta} \left[ \log \frac{\frac{\exp(-\frac{1}{2}(\theta - \mu^*(\{\lambda_i^{\bar{\zeta}}\}_{i=1}^{N^{\mathrm{cri}}}))^\top (\Sigma^*(\{\lambda_i^{\bar{\zeta}}\}_{i=1}^{N^{\mathrm{cri}}}))^{-1}(\theta - \mu^*(\{\lambda_i^{\bar{\zeta}}\}_{i=1}^{N^{\mathrm{cri}}})))}{\sqrt{|(\sigma^*(\{\lambda_i^{\bar{\zeta}}\}_{i=1}^{N^{\mathrm{cri}}}))^\top \sigma^*(\{\lambda_i^{\bar{\zeta}}\}_{i=1}^{N^{\mathrm{cri}}})|}}}{E_{\lambda_i} \left[ \frac{\exp(-\frac{1}{2}(\theta - \mu^*(\{\lambda_i^{\bar{\zeta}}\}_{i=1}^{N^{\mathrm{cri}}}))^\top (\Sigma^*(\{\lambda_i^{\bar{\zeta}}\}_{i=1}^{N^{\mathrm{cri}}}))^{-1}(\theta - \mu^*(\{\lambda_i^{\bar{\zeta}}\}_{i=1}^{N^{\mathrm{cri}}})))}{\sqrt{|(\sigma^*(\{\lambda_i^{\bar{\zeta}}\}_{i=1}^{N^{\mathrm{cri}}}))^\top \sigma^*(\{\lambda_i^{\bar{\zeta}}\}_{i=1}^{N^{\mathrm{cri}}})|}} \right]} \right],$$

$$= E_{\lambda_i, \theta} \left[ \log \frac{\exp(-\frac{1}{2}(\theta - \mu^*(\{\lambda_i^{\bar{\zeta}}\}_{i=1}^{N^{\mathrm{cri}}}))^\top (\Sigma^*(\{\lambda_i^{\bar{\zeta}}\}_{i=1}^{N^{\mathrm{cri}}}))^{-1}(\theta - \mu^*(\{\lambda_i^{\bar{\zeta}}\}_{i=1}^{N^{\mathrm{cri}}})))}{\sqrt{|(\sigma^*(\{\lambda_i^{\bar{\zeta}}\}_{i=1}^{N^{\mathrm{cri}}}))^\top \sigma^*(\{\lambda_i^{\bar{\zeta}}\}_{i=1}^{N^{\mathrm{cri}}})|}} \right]$$

$$- E_\theta \left[ \log E_{\lambda_i} \left[ \frac{\exp(-\frac{1}{2}(\theta - \mu^*(\{\lambda_i^{\bar{\zeta}}\}_{i=1}^{N^{\mathrm{cri}}}))^\top (\Sigma^*(\{\lambda_i^{\bar{\zeta}}\}_{i=1}^{N^{\mathrm{cri}}}))^{-1}(\theta - \mu^*(\{\lambda_i^{\bar{\zeta}}\}_{i=1}^{N^{\mathrm{cri}}})))}{\sqrt{|(\sigma^*(\{\lambda_i^{\bar{\zeta}}\}_{i=1}^{N^{\mathrm{cri}}}))^\top \sigma^*(\{\lambda_i^{\bar{\zeta}}\}_{i=1}^{N^{\mathrm{cri}}})|}} \right] \right],$$

$$\overset{(a)}{=} E_{\zeta \sim \mathcal{N}(0,I)} \left\{ E_{\lambda_i} \left[ \log \frac{\exp(-\frac{1}{2}\zeta^\top \zeta)}{\sqrt{|(\sigma^*(\{\lambda_i^{\bar{\zeta}}\}_{i=1}^{N^{\mathrm{cri}}}))^\top \sigma^*(\{\lambda_i^{\bar{\zeta}}\}_{i=1}^{N^{\mathrm{cri}}})|}} \right] - \log E_{\lambda_i} \left[ \frac{\exp(-\frac{1}{2}\zeta^\top \zeta)}{\sqrt{|(\sigma^*(\{\lambda_i^{\bar{\zeta}}\}_{i=1}^{N^{\mathrm{cri}}}))^\top \sigma^*(\{\lambda_i^{\bar{\zeta}_j}\}_{i=1}^{N^{\mathrm{cri}}})|}} \right] \right\},$$

$$= E_{\lambda_i} \left[ \log \frac{1}{\sqrt{|(\sigma^*(\{\lambda_i^{\bar{\zeta}}\}_{i=1}^{N^{\mathrm{cri}}}))^\top \sigma^*(\{\lambda_i^{\bar{\zeta}}\}_{i=1}^{N^{\mathrm{cri}}})|}} \right] - \log E_{\lambda_i} \left[ \frac{1}{\sqrt{|(\sigma^*(\{\lambda_i^{\bar{\zeta}_j}\}_{i=1}^{N^{\mathrm{cri}}}))^\top \sigma^*(\{\lambda_i^{\bar{\zeta}}\}_{i=1}^{N^{\mathrm{cri}}})|}} \right] \tag{10}$$

where $(a)$ follows the fact that $\theta = \mu^*(\{\lambda_i^{\bar{\zeta}}\}_{i=1}^{N^{\mathrm{cri}}}) + \sigma^*(\{\lambda_i^{\bar{\zeta}}\}_{i=1}^{N^{\mathrm{cri}}}) \circ \zeta$. Since we sample $N^{\bar{\zeta}}$ sets of mixture coefficients $\{\{\lambda_i^{\bar{\zeta}}\}_{i=1}^{N^{\mathrm{cri}}}\}_{j=1}^{N^{\bar{\zeta}}}$ from $P_{\phi_\lambda}(\lambda)$, the conditional mutual information can be estimated by

$$I(\theta; \{\bar{\mathcal{T}}_i^{\mathrm{cri}}(\lambda_i \sim P_{\phi_\lambda}(\lambda))\}_{i=1}^{N^{\mathrm{cri}}} | \{\mathcal{T}_i^{\mathrm{cri}}\}_{i=1}^{N^{\mathrm{cri}}}),$$

$$= \frac{1}{N^{\bar{\zeta}}} \sum_{j=1}^{N^{\bar{\zeta}}} \log \frac{1}{\sqrt{|(\sigma^*(\{\lambda_i^{\bar{\zeta}_j}\}_{i=1}^{N^{\mathrm{cri}}}))^\top \sigma^*(\{\lambda_i^{\bar{\zeta}_j}\}_{i=1}^{N^{\mathrm{cri}}})|}} - \log \frac{1}{N^{\bar{\zeta}}} \sum_{j=1}^{N^{\bar{\zeta}}} \frac{1}{\sqrt{|(\sigma^*(\{\lambda_i^{\bar{\zeta}_j}\}_{i=1}^{N^{\mathrm{cri}}}))^\top \sigma^*(\{\lambda_i^{\bar{\zeta}_j}\}_{i=1}^{N^{\mathrm{cri}}})|}}.$$

Therefore, we can get the gradient:

$$\nabla_{\phi_\lambda} I(\theta; \{\bar{\mathcal{T}}_i^{\mathrm{cri}}(\lambda_i \sim P_{\phi_\lambda}(\lambda))\}_{i=1}^{N^{\mathrm{cri}}} | \{\mathcal{T}_i^{\mathrm{cri}}\}_{i=1}^{N^{\mathrm{cri}}}),$$

$$= \frac{\sum_{j=1}^{N^{\bar{\zeta}}} \nabla_{\phi_\lambda} \sigma^*(\{\lambda_i^{\bar{\zeta}_j}\}_{i=1}^{N^{\mathrm{cri}}})}{|| \sum_{j=1}^{N^{\bar{\zeta}}} \sigma^*(\{\lambda_i^{\bar{\zeta}_j}\}_{i=1}^{N^{\mathrm{cri}}})||} - \frac{1}{N^{\bar{\zeta}}} \sum_{j=1}^{N^{\bar{\zeta}}} \frac{\nabla_{\phi_\lambda} \sigma^*(\{\lambda_i^{\bar{\zeta}_j}\}_{i=1}^{N^{\mathrm{cri}}})}{||\sigma^*(\{\lambda_i^{\bar{\zeta}_j}\}_{i=1}^{N^{\mathrm{cri}}})||}.$$

To get $\nabla_{\phi_\lambda}\sigma^*$, we know that $\phi^* = \arg\max E_{P_\phi(\theta)}[L(\theta, \{\bar{\mathcal{T}}_i^{\mathrm{cri}}(\lambda_i^{\bar\zeta_j})\}_{i=1}^{N^{\mathrm{cri}}}, \{\mathcal{T}_i^{\mathrm{tr}}\}_{i=1}^{N^{\mathrm{tr}}-N^{\mathrm{cri}}})]$, therefore, we have that $\nabla_\sigma E_{P_{\phi^*}(\theta)}[L(\theta, \{\bar{\mathcal{T}}_i^{\mathrm{cri}}(\lambda_i^{\bar\zeta_j})\}_{i=1}^{N^{\mathrm{cri}}}, \{\mathcal{T}_i^{\mathrm{tr}}\}_{i=1}^{N^{\mathrm{tr}}-N^{\mathrm{cri}}})] = 0$. Then we have that

$$\frac{d}{d\phi_\lambda}\nabla_\sigma E_{P_{\phi^*}(\theta)}[L(\theta, \{\bar{\mathcal{T}}_i^{\mathrm{cri}}(\lambda_i^{\bar\zeta_j})\}_{i=1}^{N^{\mathrm{cri}}}, \{\mathcal{T}_i^{\mathrm{tr}}\}_{i=1}^{N^{\mathrm{tr}}-N^{\mathrm{cri}}})],$$

$$= \nabla_{\sigma\phi_\lambda} E_{P_{\phi^*}(\theta)}[L(\theta, \{\bar{\mathcal{T}}_i^{\mathrm{cri}}(\lambda_i^{\bar\zeta_j})\}_{i=1}^{N^{\mathrm{cri}}}, \{\mathcal{T}_i^{\mathrm{tr}}\}_{i=1}^{N^{\mathrm{tr}}-N^{\mathrm{cri}}})]$$

$$+ \nabla_{\sigma\sigma} E_{P_{\phi^*}(\theta)}[L(\theta, \{\bar{\mathcal{T}}_i^{\mathrm{cri}}(\lambda_i^{\bar\zeta_j})\}_{i=1}^{N^{\mathrm{cri}}}, \{\mathcal{T}_i^{\mathrm{tr}}\}_{i=1}^{N^{\mathrm{tr}}-N^{\mathrm{cri}}})]\nabla_{\phi_\lambda}\sigma^* = 0,$$

$$\Rightarrow \nabla_{\phi_\lambda}\sigma^* = -\Big[\nabla_{\sigma\sigma}^2 E_{P_{\phi^*}(\theta)}[L(\theta, \{\bar{\mathcal{T}}_i^{\mathrm{cri}}(\lambda_i^{\bar\zeta_j})\}_{i=1}^{N^{\mathrm{cri}}}, \{\mathcal{T}_i^{\mathrm{tr}}\}_{i=1}^{N^{\mathrm{tr}}-N^{\mathrm{cri}}})]\Big]^{-1}.$$

$$\nabla_{\sigma\phi_\lambda}^2 E_{P_{\phi^*}(\theta)}[L(\theta, \{\bar{\mathcal{T}}_i^{\mathrm{cri}}(\lambda_i^{\bar\zeta_j})\}_{i=1}^{N^{\mathrm{cri}}}, \{\mathcal{T}_i^{\mathrm{tr}}\}_{i=1}^{N^{\mathrm{tr}}-N^{\mathrm{cri}}})].$$

# F GRADIENTS

This section provides all the gradients needed in this paper.

## F.1 META-GRADIENTS FOR MAJOR META-RL METHODS

Recall the problem formulation (1) of meta-RL as follows where we omit the superscript for simplicity:

$$\max_\theta\ L(\theta, \{\mathcal{T}_i\}_{i=1}^N) = \frac{1}{N}\sum_{i=1}^N J_i(\pi_i(\theta)), \quad \text{s.t. } \pi_i(\theta) = Alg(\pi_\theta, \mathcal{T}_i).$$

The meta-gradient is the gradient of the upper-level objective w.r.t. $\theta$, i.e., $\nabla_\theta L(\theta, \{\mathcal{T}_i\}_{i=1}^N)$. The meta-gradient is different for different algorithms because different algorithms use different ways to compute the task specific adaptations $\pi_i(\theta)$. Here, we provide the meta-gradients for several major meta-RL algorithms, including MAML (Finn et al., 2017; Fallah et al., 2021), iMAML (Rajeswaran et al., 2019), and context-based meta-RL (e.g., CAVIA (Zintgraf et al., 2019)).

**Lemma 4.** *The meta-gradients for MAML, iMAML, and CAVIA are respectively:*

$$\nabla_\theta L(\theta, \{\mathcal{T}_i\}_{i=1}^N) = \frac{1}{N}\sum_{i=1}^N [I + \alpha\nabla_{\theta\theta}^2 J_i(\pi_\theta)]\nabla_{\theta_i} J_i(\pi_{\theta_i}), \qquad (MAML)$$

$$\nabla_\theta L(\theta, \{\mathcal{T}_i\}_{i=1}^N) = \frac{1}{N}\sum_{i=1}^N [1 + \frac{1}{\bar\lambda}\nabla_{\psi\psi}^2 J_i(\pi_{\theta_i'})]^{-1}\nabla_{\theta_i} J_i(\pi_{\theta_i}), \qquad (iMAML)$$

$$\nabla_\theta L(\theta, \{\mathcal{T}_i\}_{i=1}^N) = \frac{1}{N}\sum_{i=1}^N \nabla_\theta J_i(\pi_\theta(\cdot|\cdot, \psi_i'')), \qquad (CAVIA)$$

*where $\alpha$ is a step size, $\theta_i = \theta + \alpha\nabla_\theta J_i(\pi_\theta)$, $\nabla_{\theta_i} J_i(\pi_{\theta_i}) = E_{(s,a)\sim\rho^{\pi_{\theta_i}}}[\nabla_{\theta_i}\log\pi_{\theta_i}(a|s)A_i^{\pi_{\theta_i}}(s,a)]$, $\nabla_{\theta\theta}^2 J_i(\pi_\theta) = E_{(s,a)\sim\rho^{\pi_\theta}}\Big[\sum_{t=0}^\infty \gamma^t \nabla_\theta E_{(s,a)\sim\rho^{\pi_\theta}}[\log\pi_\theta(a|s)Q_i^{\pi_\theta}(s,a)](\nabla_\theta\log\pi_\theta(a|s))^\top + \nabla_{\theta\theta}^2 E_{(s,a)\sim\rho^{\pi_\theta}}[\log\pi_\theta(a|s)Q_i^{\pi_\theta}(s,a)]\Big]$, $\bar\lambda$ is a hyper-parameter, $\theta_i' = \arg\max_\psi J_i(\pi_\psi) + \frac{\bar\lambda}{2}||\psi - \theta||^2$, $\pi_\theta(\cdot|\cdot, \psi_i'')$ is a context-based policy where $\psi_i'' = \psi_0 + \alpha\nabla_\psi J_i(\pi_\theta(\cdot|\cdot, \psi_0))$ is the context.*

*Proof.* MAML computes the task-specific adaptation via one-step gradient ascent. In specific, suppose the task-specific adaptation is $\pi_{\theta_i} = \pi_i(\theta)$, and thus $\theta_i = \theta + \alpha\nabla_\theta J_i(\pi_\theta)$. Therefore, the meta-gradient is $\nabla_\theta L(\theta, \{\mathcal{T}_i\}_{i=1}^N) = \frac{1}{N}\sum_{i=1}^N \nabla_\theta J_i(\pi_{\theta_i}) = \frac{1}{N}\sum_{i=1}^N (\nabla_\theta\theta_i)^\top \nabla_{\theta_i} J_i(\pi_{\theta_i}) = \frac{1}{N}\sum_{i=1}^N [I + \alpha\nabla_{\theta\theta}^2 J_i(\pi_\theta)]\nabla_{\theta_i} J_i(\pi_{\theta_i})$. From (Fallah et al., 2021), we can get that the policy gradient is $\nabla_{\theta_i} J_i(\pi_{\theta_i}) = E_{(s,a)\sim\rho^{\pi_{\theta_i}}}[\nabla_{\theta_i}\log\pi_{\theta_i}(a|s)A_i^{\pi_{\theta_i}}(s,a)]$ and the Hessian is $\nabla_{\theta\theta}^2 J_i(\pi_\theta) = E_{(s,a)\sim\rho^{\pi_\theta}}\Big[\sum_{t=0}^\infty \gamma^t \nabla_\theta E_{(s,a)\sim\rho^{\pi_\theta}}[\log\pi_\theta(a|s)Q_i^{\pi_\theta}(s,a)](\nabla_\theta\log\pi_\theta(a|s))^\top + \nabla_{\theta\theta}^2 E_{(s,a)\sim\rho^{\pi_\theta}}[\log\pi_\theta(a|s)Q_i^{\pi_\theta}(s,a)]\Big]$.

iMAML solves the optimization problem to get the task-specific adaptation $\pi_{\theta_i'}$ such that $\theta_i' = \arg\max_\psi J_i(\pi_\psi) + \frac{\bar{\lambda}}{2}||\psi - \theta||^2$ where $\bar{\lambda}$ is a hyper-parameter. Since $\theta_i'$ is the optimal parameter of the problem $\max_\psi J_i(\pi_\psi) + \frac{\bar{\lambda}}{2}||\psi - \theta||^2$, we know that $\nabla_\psi J_i(\pi_{\theta_i'}) + \bar{\lambda}(\theta_i' - \theta) = 0$. Take gradient w.r.t. $\theta$ on both sides, we can get that $(\nabla_\theta \theta_i')^\top \nabla_{\psi\psi}^2 J_i(\pi_{\theta_i'}) + \bar{\lambda}(\nabla_\theta \theta_i' - I) = 0 \Rightarrow \nabla_\theta \theta_i' = [1 + \frac{1}{\bar{\lambda}}\nabla_{\psi\psi}^2 J_i(\pi_{\theta_i'})]^{-1}$. Therefore, the meta-gradient is $\nabla_\theta L(\theta, \{\mathcal{T}_i\}_{i=1}^N) = \frac{1}{N}\sum_{i=1}^N \nabla_\theta J_i(\pi_{\theta_i}) = \frac{1}{N}\sum_{i=1}^N (\nabla_\theta \theta_i)^\top \nabla_{\theta_i} J_i(\pi_{\theta_i}) = \frac{1}{N}\sum_{i=1}^N [1 + \frac{1}{\bar{\lambda}}\nabla_{\psi\psi}^2 J_i(\pi_{\theta_i'})]^{-1}\nabla_{\theta_i} J_i(\pi_{\theta_i})$.

CAVIA learns a context-based policy $\pi_\theta(a|s, \psi_i'')$ and uses MAML-like method to update $\psi_i'' = \psi_0 + \alpha\nabla_\psi J_i(\pi_\theta(\cdot|\cdot, \psi_0))$. Therefore, the meta-gradient is $\nabla_\theta L(\theta, \{\mathcal{T}_i\}_{i=1}^N) = \frac{1}{N}\sum_{i=1}^N \nabla_\theta J_i(\pi_\theta(\cdot|\cdot, \psi_i''))$. $\square$

### F.2 OTHER GRADIENTS

This part provides the expressions of $\nabla_{\sigma\sigma}^2 E_{P_{\phi^*}(\theta)}[L(\theta, \{\bar{\mathcal{T}}_i^{\text{cri}}(\lambda_i^{\bar{\zeta}_j})\}_{i=1}^{N^{\text{cri}}}, \{\mathcal{T}_i^{\text{tr}}\}_{i=1}^{N^{\text{tr}}-N^{\text{cri}}})]$ and $\nabla_{\sigma\phi_\lambda}^2 E_{P_{\phi^*}(\theta)}[L(\theta, \{\bar{\mathcal{T}}_i^{\text{cri}}(\lambda_i^{\bar{\zeta}_j})\}_{i=1}^{N^{\text{cri}}}, \{\mathcal{T}_i^{\text{tr}}\}_{i=1}^{N^{\text{tr}}-N^{\text{cri}}})]$ needed in Lemma 1.

**Lemma 5.** *We have the following expressions:*

$$\nabla_{\sigma\sigma}^2 E_{P_{\phi^*}(\theta)}[L(\theta, \{\bar{\mathcal{T}}_i^{\text{cri}}(\lambda_i^{\bar{\zeta}_j})\}_{i=1}^{N^{\text{cri}}}, \{\mathcal{T}_i^{\text{tr}}\}_{i=1}^{N^{\text{tr}}-N^{\text{cri}}})],$$

$$= E_{\zeta\sim\mathcal{N}(0,I)}\Big[\frac{1}{N^{tr}}[\sum_{i=1}^{N^{cri}} \nabla_{\sigma\sigma}^2 \bar{J}_i^{cri}(\pi_i^{cri}(\mu^* + \sigma^* \circ \zeta), \lambda_i^{\bar{\zeta}_j}) + \sum_{i=1}^{N^{tr}-N^{cri}} \nabla_{\sigma\sigma}^2 J_i^{tr}(\pi_i^{tr}(\mu^* + \sigma^* \circ \zeta))]\Big],$$

$$\nabla_{\sigma\phi_\lambda}^2 E_{P_{\phi^*}(\theta)}[L(\theta, \{\bar{\mathcal{T}}_i^{\text{cri}}(\lambda_i^{\bar{\zeta}_j})\}_{i=1}^{N^{\text{cri}}}, \{\mathcal{T}_i^{\text{tr}}\}_{i=1}^{N^{\text{tr}}-N^{\text{cri}}})],$$

$$= E_{\zeta\sim\mathcal{N}(0,I)}\Big[\frac{1}{N^{tr}}[\sum_{i=1}^{N^{cri}} \nabla_{\phi_\lambda}\lambda_j \cdot$$

$$\int_{(s_{jj'}, a_{jj'})\in\mathcal{S}\times\mathcal{A}} \Big[\bar{\rho}^{\pi_{\theta_i}, \lambda_j}(s_{jj'}, a_{jj'})\Big(\nabla_{\theta_i s} \log\pi_{\theta_i}(\bar{a}_{jj'}|\bar{s}_{jj'})(s_j - s_{j'})\Big)\bar{A}_{jj'}$$

$$+ \bar{\rho}^{\pi_{\theta_i}, \lambda_j}(s_{jj'}, a_{jj'})\nabla_{\theta_i} \log\pi_{\theta_i}(\bar{a}_{jj'}|\bar{s}_{jj'})(A_i^{\pi_{\theta_i}}(s_j, a_j) - A_i^{\pi_{\theta_i}}(s_{j'}, a_{j'})))\Big]da_{jj'}ds_{jj'}\Big],$$

*where the expression of the second-order term* $\nabla_{\sigma\sigma}^2 \bar{J}_i^{cri}(\pi_i^{cri}(\mu^* + \sigma^* \circ \zeta), \lambda_i)$ *can be found in Lemma 4.*

*Proof.* Recall that $\phi^* = (\mu^*, \sigma^*)$, $\theta = \mu + \sigma \circ \zeta$, and $\zeta \sim \mathcal{N}(0, I)$. Therefore, we have that

$$\nabla_\sigma E_{P_{\phi^*}(\theta)}[L(\theta, \{\bar{\mathcal{T}}_i^{\text{cri}}(\lambda_i^{\bar{\zeta}_j})\}_{i=1}^{N^{\text{cri}}}, \{\mathcal{T}_i^{\text{tr}}\}_{i=1}^{N^{\text{tr}}-N^{\text{cri}}})],$$

$$= E_{\zeta\sim\mathcal{N}(0,I)}[\nabla_\sigma L(\mu^* + \sigma^* \circ \zeta, \{\bar{\mathcal{T}}_i^{\text{cri}}(\lambda_i^{\bar{\zeta}_j})\}_{i=1}^{N^{\text{cri}}}, \{\mathcal{T}_i^{\text{tr}}\}_{i=1}^{N^{\text{tr}}-N^{\text{cri}}})],$$

$$= E_{\zeta\sim\mathcal{N}(0,I)}\Big[\frac{1}{N^{tr}}[\sum_{i=1}^{N^{cri}} \nabla_\sigma \bar{J}_i^{cri}(\pi_i^{cri}(\mu^* + \sigma^* \circ \zeta), \lambda_i) + \sum_{i=1}^{N^{tr}-N^{cri}} \nabla_\sigma J_i^{tr}(\pi_i^{tr}(\mu^* + \sigma^* \circ \zeta))]\Big]. \quad (11)$$

Therefore, we can get the Hessian:

$$\nabla_{\sigma\sigma}^2 E_{P_{\phi^*}(\theta)}[L(\theta, \{\bar{\mathcal{T}}_i^{\text{cri}}(\lambda_i^{\bar{\zeta}_j})\}_{i=1}^{N^{\text{cri}}}, \{\mathcal{T}_i^{\text{tr}}\}_{i=1}^{N^{\text{tr}}-N^{\text{cri}}})],$$

$$= E_{\zeta\sim\mathcal{N}(0,I)}\Big[\frac{1}{N^{tr}}[\sum_{i=1}^{N^{cri}} \nabla_{\sigma\sigma}^2 \bar{J}_i^{cri}(\pi_i^{cri}(\mu^* + \sigma^* \circ \zeta), \lambda_i^{\bar{\zeta}_j}) + \sum_{i=1}^{N^{tr}-N^{cri}} \nabla_{\sigma\sigma}^2 J_i^{tr}(\pi_i^{tr}(\mu^* + \sigma^* \circ \zeta))]\Big],$$

where the expression of the second-order term $\nabla_{\sigma\sigma}^2 \bar{J}_i^{cri}(\pi_i^{cri}(\mu^* + \sigma^* \circ \zeta), \lambda_i)$ can be found in Lemma 4. Similarly, we can get that

$$\nabla_{\sigma\phi_\lambda}^2 E_{P_{\phi^*}(\theta)}[L(\theta, \{\bar{\mathcal{T}}_i^{\text{cri}}(\lambda_i^{\bar{\zeta}_j})\}_{i=1}^{N^{\text{cri}}}, \{\mathcal{T}_i^{\text{tr}}\}_{i=1}^{N^{\text{tr}}-N^{\text{cri}}})],$$

$$= E_{\zeta\sim\mathcal{N}(0,I)}\Big[\frac{1}{N^{tr}}[\sum_{i=1}^{N^{cri}} \nabla_{\sigma\phi_\lambda}^2 \bar{J}_i^{cri}(\pi_i^{cri}(\mu^* + \sigma^* \circ \zeta), \lambda_i^{\bar{\zeta}_j}) + \sum_{i=1}^{N^{tr}-N^{cri}} \nabla_{\sigma\phi_\lambda}^2 J_i^{tr}(\pi_i^{tr}(\mu^* + \sigma^* \circ \zeta))]\Big],$$

$$= E_{\zeta \sim \mathcal{N}(0,I)} \Big[ \frac{1}{N^{\text{tr}}} [\sum_{i=1}^{N^{\text{cri}}} \nabla^2_{\sigma \phi_\lambda} \bar{J}_i^{\text{cri}}(\pi_i^{\text{cri}}(\mu^* + \sigma^* \circ \zeta), \mu_\lambda + \sigma_\lambda \bar{\zeta}_j) \Big].$$

Now we need to derive the expression of $\nabla^2_{\sigma \phi_\lambda} \bar{J}_i^{\text{cri}}(\pi_i^{\text{cri}}(\mu^* + \sigma^* \circ \zeta), \mu_\lambda + \sigma_\lambda \bar{\zeta}_j)$. Suppose we use MAML, and thus the first-order gradient $\nabla_\sigma \bar{J}_i^{\text{cri}}(\pi_i^{\text{cri}}(\mu^* + \sigma^* \circ \zeta), \mu_\lambda + \sigma_\lambda \bar{\zeta}_j) = [I + \alpha \nabla^2_{\sigma\sigma} \bar{J}_i(\pi_{\mu^* + \sigma^* \circ \zeta}, \mu_\lambda + \sigma_\lambda \bar{\zeta}_j)] \nabla_{\theta_i} \bar{J}_i(\pi_{\theta_i}, \mu_\lambda + \sigma_\lambda \bar{\zeta}_j)]$ where $\theta_i = \mu^* + \sigma^* \circ \zeta + \alpha \nabla_\theta \bar{J}_i(\pi_{\mu^* + \sigma^* \circ \zeta}, \mu_\lambda + \sigma_\lambda \bar{\zeta}_j)$ and $\theta = \mu^* + \sigma^* \circ \zeta$. Following the first-order MAML method in (Fallah et al., 2020), we use the gradient $\nabla_\sigma \bar{J}_i^{\text{cri}}(\pi_i^{\text{cri}}(\mu^* + \sigma^* \circ \zeta), \mu_\lambda + \sigma_\lambda \bar{\zeta}_j) = \nabla_\sigma \bar{J}_i(\pi_{\theta_i}, \mu_\lambda + \sigma_\lambda \bar{\zeta}_j)]$. To get the term $\nabla^2_{\sigma \phi_\lambda} \bar{J}_i^{\text{cri}}(\pi_i^{\text{cri}}(\mu^* + \sigma^* \circ \zeta), \mu_\lambda + \sigma_\lambda \bar{\zeta}_j)$, we derive $\nabla_{\theta_i \phi_\lambda} \bar{J}_i(\pi_{\theta_i}, \mu_\lambda + \sigma_\lambda \bar{\zeta}_j)$.

$$\nabla^2_{\phi_\lambda, \theta_i} \bar{J}_i(\pi_{\theta_i}, \mu_\lambda + \sigma_\lambda \bar{\zeta}_j) = \nabla_{\phi_\lambda} E_{(s_{jj'}, a_{jj'}) \sim \bar{\rho}^{\pi_{\theta_i}, \lambda_j}} [\nabla_{\theta_i} \log \pi_{\theta_i}(\bar{a}_{jj'} | \bar{s}_{jj'}) \bar{A}_{jj'}],$$

$$= \nabla_{\phi_\lambda} \int_{(s_{jj'}, a_{jj'}) \in \mathcal{S} \times \mathcal{A}} \bar{\rho}^{\pi_{\theta_i}, \lambda_j}(s_{jj'}, a_{jj'}) \nabla_{\theta_i} \log \pi_{\theta_i}(\bar{a}_{jj'} | \bar{s}_{jj'}) \bar{A}_{jj'} da_{jj'} ds_{jj'},$$

$$= \nabla_{\phi_\lambda} \int_{(s_{jj'}, a_{jj'}) \in \mathcal{S} \times \mathcal{A}} \Big[ \bar{\rho}^{\pi_{\theta_i}, \lambda_j}(s_{jj'}, a_{jj'}) \nabla_{\theta_i} \log \pi_{\theta_i}(\bar{a}_{jj'} | \bar{s}_{jj'}) \bar{A}_{jj'} \Big] da_{jj'} ds_{jj'},$$

$$= \nabla_{\phi_\lambda} \lambda_j \cdot \int_{(s_{jj'}, a_{jj'}) \in \mathcal{S} \times \mathcal{A}} \nabla_{\lambda_j} \Big[ \bar{\rho}^{\pi_{\theta_i}, \lambda_j}(s_{jj'}, a_{jj'}) \nabla_{\theta_i} \log \pi_{\theta_i}(\bar{a}_{jj'} | \bar{s}_{jj'}) \bar{A}_{jj'} \Big] da_{jj'} ds_{jj'},$$

$$\overset{(a)}{=} \nabla_{\phi_\lambda} \lambda_j \cdot \int_{(s_{jj'}, a_{jj'}) \in \mathcal{S} \times \mathcal{A}} \Big[ \bar{\rho}^{\pi_{\theta_i}, \lambda_j}(s_{jj'}, a_{jj'}) \Big( \nabla_{\theta_i s} \log \pi_{\theta_i}(\bar{a}_{jj'} | \bar{s}_{jj'})(s_j - s_{j'}) \Big) \bar{A}_{jj'}$$

$$+ \bar{\rho}^{\pi_{\theta_i}, \lambda_j}(s_{jj'}, a_{jj'}) \nabla_{\theta_i} \log \pi_{\theta_i}(\bar{a}_{jj'} | \bar{s}_{jj'})(A_i^{\pi_{\theta_i}}(s_j, a_j) - A_i^{\pi_{\theta_i}}(s_{j'}, a_{j'})) \Big] da_{jj'} ds_{jj'},$$

where $(a)$ follows the fact that $\nabla_{\lambda_i} \bar{\rho}^{\pi_{\theta_i}, \lambda_j}(s_{jj'}, a_{jj'}) = 0$ and $\nabla_{\theta a} \log \pi_{\theta_i}(\bar{a}_{jj'} | \bar{s}_{jj'})$ because they include indicator functions. Therefore, we have that

$$\nabla^2_{\sigma \phi_\lambda} E_{P_{\phi^*}(\theta)}[L(\theta, \{\bar{\mathcal{T}}_i^{\text{cri}}(\lambda_i^{\bar{\zeta}_j})\}_{i=1}^{N^{\text{cri}}}, \{\mathcal{T}_i^{\text{tr}}\}_{i=1}^{N^{\text{tr}} - N^{\text{cri}}})],$$

$$= E_{\zeta \sim \mathcal{N}(0,I)} \Big[ \frac{1}{N^{\text{tr}}} [\sum_{i=1}^{N^{\text{cri}}} \nabla_{\phi_\lambda} \lambda_j \cdot$$

$$\int_{(s_{jj'}, a_{jj'}) \in \mathcal{S} \times \mathcal{A}} \Big[ \bar{\rho}^{\pi_{\theta_i}, \lambda_j}(s_{jj'}, a_{jj'}) \Big( \nabla_{\theta_i s} \log \pi_{\theta_i}(\bar{a}_{jj'} | \bar{s}_{jj'})(s_j - s_{j'}) \Big) \bar{A}_{jj'}$$

$$+ \bar{\rho}^{\pi_{\theta_i}, \lambda_j}(s_{jj'}, a_{jj'}) \nabla_{\theta_i} \log \pi_{\theta_i}(\bar{a}_{jj'} | \bar{s}_{jj'})(A_i^{\pi_{\theta_i}}(s_j, a_j) - A_i^{\pi_{\theta_i}}(s_{j'}, a_{j'}))) \Big] da_{jj'} ds_{jj'} \Big].$$

$$\square$$

## G  PROOF OF THEOREM 1

This section first prove that the conditional mutual information $I(\theta; \{\bar{\mathcal{T}}_i^{\text{cri}}(\lambda_i \sim P_{\phi_\lambda}(\lambda))\}_{i=1}^{N^{\text{cri}}} | \{\mathcal{T}_i^{\text{cri}}\}_{i=1}^{N^{\text{cri}}})$ is $C_I$-Lipschitz continuous and $\bar{C}_I$-smooth where $C_I$ and $\bar{C}_I$ are positive constants in Claim 1, and then prove that Algorithm 1 converges at the rate of $O(1/\sqrt{K})$.

**Claim 1.** *The conditional mutual information is $C_I$-Lipschitz continuous and $\bar{C}_I$-smooth where $C_I$ and $\bar{C}_I$ are positive constants.*

*Proof.* From (10), we know that

$$I(\theta; \{\bar{\mathcal{T}}_i^{\text{cri}}(\lambda_i \sim P_{\phi_\lambda}(\lambda))\}_{i=1}^{N^{\text{cri}}} | \{\mathcal{T}_i^{\text{cri}}\}_{i=1}^{N^{\text{cri}}}),$$

$$= E_{\lambda_i} \Big[ \log \frac{1}{\sqrt{|(\sigma^*(\{\lambda_i^{\bar{\zeta}}\}_{i=1}^{N^{\text{cri}}}))^\top \sigma^*(\{\lambda_i^{\bar{\zeta}}\}_{i=1}^{N^{\text{cri}}})|}} \Big] - \log E_{\lambda_i} \Big[ \frac{1}{\sqrt{|(\sigma^*(\{\lambda_i^{\bar{\zeta}}\}_{i=1}^{N^{\text{cri}}}))^\top \sigma^*(\{\lambda_i^{\bar{\zeta}}\}_{i=1}^{N^{\text{cri}}})|}} \Big],$$

where $\lambda_i^{\bar{\zeta}} = \mu_\lambda + \sigma_\lambda \bar{\zeta}_i$ and $\bar{\zeta}_i \sim \mathcal{N}(0,1)$. Therefore, we can get the gradient

$$\nabla_{\phi_\lambda} I(\theta; \{\bar{\mathcal{T}}_i^{\text{cri}}(\lambda_i \sim P_{\phi_\lambda}(\lambda))\}_{i=1}^{N^{\text{cri}}} | \{\mathcal{T}_i^{\text{cri}}\}_{i=1}^{N^{\text{cri}}}),$$

$$= \frac{E_{\bar{\zeta}\sim\mathcal{N}(0,1)}[\nabla_{\phi_\lambda}\sigma^*(\{\lambda_i^{\bar{\zeta}}\}_{i=1}^{N^{\mathrm{cri}}})]}{||E_{\bar{\zeta}\sim\mathcal{N}(0,1)}[\sigma^*(\{\lambda_i^{\bar{\zeta}}\}_{i=1}^{N^{\mathrm{cri}}})]||} - E_{\bar{\zeta}\sim\mathcal{N}(0,1)}\Big[\frac{\nabla_{\phi_\lambda}\sigma^*(\{\lambda_i^{\bar{\zeta}}\}_{i=1}^{N^{\mathrm{cri}}})}{||\sigma^*(\{\lambda_i^{\bar{\zeta}}\}_{i=1}^{N^{\mathrm{cri}}})||}\Big]. \tag{12}$$

Now, we consider the Hessian

$$\nabla^2_{\phi_\lambda\phi_\lambda} I(\theta; \{\bar{\mathcal{T}}_i^{\mathrm{cri}}(\lambda_i \sim P_{\phi_\lambda}(\lambda))\}_{i=1}^{N^{\mathrm{cri}}}|\{\mathcal{T}_i^{\mathrm{cri}}\}_{i=1}^{N^{\mathrm{cri}}}),$$

$$= \nabla_{\phi_\lambda} \frac{E_{\bar{\zeta}\sim\mathcal{N}(0,1)}[\nabla_{\phi_\lambda}\sigma^*(\{\lambda_i^{\bar{\zeta}}\}_{i=1}^{N^{\mathrm{cri}}})]}{||E_{\bar{\zeta}\sim\mathcal{N}(0,1)}[\sigma^*(\{\lambda_i^{\bar{\zeta}}\}_{i=1}^{N^{\mathrm{cri}}})]||} - E_{\bar{\zeta}\sim\mathcal{N}(0,1)}\Big[\nabla_{\phi_\lambda}\Big[\frac{\nabla_{\phi_\lambda}\sigma^*(\{\lambda_i^{\bar{\zeta}}\}_{i=1}^{N^{\mathrm{cri}}})}{||\sigma^*(\{\lambda_i^{\bar{\zeta}}\}_{i=1}^{N^{\mathrm{cri}}})||}\Big]\Big],,$$

$$= \frac{E_{\bar{\zeta}\sim\mathcal{N}(0,1)}[\nabla^2_{\phi_\lambda\phi_\lambda}\sigma^*(\{\lambda_i^{\bar{\zeta}}\}_{i=1}^{N^{\mathrm{cri}}})]}{||E_{\bar{\zeta}\sim\mathcal{N}(0,1)}[\sigma^*(\{\lambda_i^{\bar{\zeta}}\}_{i=1}^{N^{\mathrm{cri}}})]||}$$

$$- \frac{E_{\bar{\zeta}\sim\mathcal{N}(0,1)}[\nabla_{\phi_\lambda}\sigma^*(\{\lambda_i^{\bar{\zeta}}\}_{i=1}^{N^{\mathrm{cri}}})](E_{\bar{\zeta}\sim\mathcal{N}(0,1)}[\sigma^*(\{\lambda_i^{\bar{\zeta}}\}_{i=1}^{N^{\mathrm{cri}}})])^\top E_{\bar{\zeta}\sim\mathcal{N}(0,1)}[\nabla_{\phi_\lambda}\sigma^*(\{\lambda_i^{\bar{\zeta}}\}_{i=1}^{N^{\mathrm{cri}}})]}{||E_{\bar{\zeta}\sim\mathcal{N}(0,1)}[\sigma^*(\{\lambda_i^{\bar{\zeta}}\}_{i=1}^{N^{\mathrm{cri}}})]||^3}$$

$$- E_{\bar{\zeta}\sim\mathcal{N}(0,1)}\Big[\frac{\nabla^2_{\phi_\lambda\phi_\lambda}\sigma^*(\{\lambda_i^{\bar{\zeta}}\}_{i=1}^{N^{\mathrm{cri}}})}{||\sigma^*(\{\lambda_i^{\bar{\zeta}}\}_{i=1}^{N^{\mathrm{cri}}})||} - \frac{\nabla_{\phi_\lambda}\sigma^*(\{\lambda_i^{\bar{\zeta}}\}_{i=1}^{N^{\mathrm{cri}}})(\sigma^*(\{\lambda_i^{\bar{\zeta}}\}_{i=1}^{N^{\mathrm{cri}}}))^\top \nabla_{\phi_\lambda}\sigma^*(\{\lambda_i^{\bar{\zeta}}\}_{i=1}^{N^{\mathrm{cri}}})}{||\sigma^*(\{\lambda_i^{\bar{\zeta}}\}_{i=1}^{N^{\mathrm{cri}}})||^3}\Big]. \tag{13}$$

From (12), we know that if we can lower bound $||\sigma^*||$ and upper bound $||\nabla_{\phi_\lambda}\sigma^*||$, the norm of the gradient $\nabla_{\phi_\lambda} I(\theta; \{\bar{\mathcal{T}}_i^{\mathrm{cri}}(\lambda_i \sim P_{\phi_\lambda}(\lambda))\}_{i=1}^{N^{\mathrm{cri}}}|\{\mathcal{T}_i^{\mathrm{cri}}\}_{i=1}^{N^{\mathrm{cri}}})$ is bounded. From (13), we know that if we can lower bound $||\sigma^*||$ and upper bound $||\nabla_{\phi_\lambda}\sigma^*||$ and $||\nabla^2_{\phi_\lambda\phi_\lambda}\sigma^*||$, the norm of the Hessian $||\nabla^2_{\phi_\lambda\phi_\lambda} I(\theta; \{\bar{\mathcal{T}}_i^{\mathrm{cri}}(\lambda_i \sim P_{\phi_\lambda}(\lambda))\}_{i=1}^{N^{\mathrm{cri}}}|\{\mathcal{T}_i^{\mathrm{cri}}\}_{i=1}^{N^{\mathrm{cri}}})||$ is bounded. Note that $\lambda \in [0,1]$ is bounded within a compact set. Therefore, as long as we can prove that $\sigma^*$, $\nabla_{\phi_\lambda}\sigma^*$, and $\nabla^2_{\phi_\lambda\phi_\lambda}\sigma^*$ are continuous in $\lambda$, their norms are both upper bounded and lower bounded. To show that $\sigma^*$, $\nabla_{\phi_\lambda}\sigma^*$, and $\nabla^2_{\phi_\lambda\phi_\lambda}\sigma^*$ are continuous in $\lambda$, we can show that they are differentiable w.r.t. $\lambda$. Since $\phi_\lambda$ is differentiable w.r.t. $\lambda$, we only need to show that $\sigma^*$, $\nabla_{\phi_\lambda}\sigma^*$, and $\nabla^2_{\phi_\lambda\phi_\lambda}\sigma^*$ are differentiable w.r.t. $\phi_\lambda$. This suffices to show that $\nabla_{\phi_\lambda}\sigma^*$, $\nabla^2_{\phi_\lambda\phi_\lambda}\sigma^*$, and $\nabla^3_{\phi_\lambda\phi_\lambda\phi_\lambda}\sigma^*$ exist.

From Lemma 1, we know that $\nabla_{\phi_\lambda}\sigma^*$ exists and

$$\nabla_{\phi_\lambda}\sigma^* = -\Big[\nabla^2_{\sigma\sigma} E_{P_{\phi^*}(\theta)}[L(\theta, \{\bar{\mathcal{T}}_i^{\mathrm{cri}}(\lambda_i^{\bar{\zeta}_j})\}_{i=1}^{N^{\mathrm{cri}}}, \{\mathcal{T}_i^{\mathrm{tr}}\}_{i=1}^{N^{\mathrm{tr}}-N^{\mathrm{cri}}})]\Big]^{-1}.$$
$$\nabla^2_{\sigma\phi_\lambda} E_{P_{\phi^*}(\theta)}[L(\theta, \{\bar{\mathcal{T}}_i^{\mathrm{cri}}(\lambda_i^{\bar{\zeta}_j})\}_{i=1}^{N^{\mathrm{cri}}}, \{\mathcal{T}_i^{\mathrm{tr}}\}_{i=1}^{N^{\mathrm{tr}}-N^{\mathrm{cri}}})].$$

Since $\log \pi_\theta$ is smooth in $\theta$ (Assumption 1), we can see that $L(\theta, \{\bar{\mathcal{T}}_i^{\mathrm{cri}}(\lambda_i^{\bar{\zeta}_j})\}_{i=1}^{N^{\mathrm{cri}}}, \{\mathcal{T}_i^{\mathrm{tr}}\}_{i=1}^{N^{\mathrm{tr}}-N^{\mathrm{cri}}})$ is also smooth in $\theta$. Since $\theta$ is smooth in $\sigma$, $L(\theta, \{\bar{\mathcal{T}}_i^{\mathrm{cri}}(\lambda_i^{\bar{\zeta}_j})\}_{i=1}^{N^{\mathrm{cri}}}, \{\mathcal{T}_i^{\mathrm{tr}}\}_{i=1}^{N^{\mathrm{tr}}-N^{\mathrm{cri}}})$ is also smooth in $\sigma$. Similarly, we can derive

$$\nabla^2_{\phi_\lambda\phi_\lambda}\sigma^* = \Big[\nabla^2_{\sigma\sigma} E_{P_{\phi^*}(\theta)}[L(\theta, \{\bar{\mathcal{T}}_i^{\mathrm{cri}}(\lambda_i^{\bar{\zeta}_j})\}_{i=1}^{N^{\mathrm{cri}}}, \{\mathcal{T}_i^{\mathrm{tr}}\}_{i=1}^{N^{\mathrm{tr}}-N^{\mathrm{cri}}})]\Big]^{-1}.$$
$$\nabla^3_{\sigma\sigma\phi_\lambda} E_{P_{\phi^*}(\theta)}[L(\theta, \{\bar{\mathcal{T}}_i^{\mathrm{cri}}(\lambda_i^{\bar{\zeta}_j})\}_{i=1}^{N^{\mathrm{cri}}}, \{\mathcal{T}_i^{\mathrm{tr}}\}_{i=1}^{N^{\mathrm{tr}}-N^{\mathrm{cri}}})].$$
$$\Big[\nabla^2_{\sigma\sigma} E_{P_{\phi^*}(\theta)}[L(\theta, \{\bar{\mathcal{T}}_i^{\mathrm{cri}}(\lambda_i^{\bar{\zeta}_j})\}_{i=1}^{N^{\mathrm{cri}}}, \{\mathcal{T}_i^{\mathrm{tr}}\}_{i=1}^{N^{\mathrm{tr}}-N^{\mathrm{cri}}})]\Big]^{-1}$$
$$- \Big[\nabla^2_{\sigma\sigma} E_{P_{\phi^*}(\theta)}[L(\theta, \{\bar{\mathcal{T}}_i^{\mathrm{cri}}(\lambda_i^{\bar{\zeta}_j})\}_{i=1}^{N^{\mathrm{cri}}}, \{\mathcal{T}_i^{\mathrm{tr}}\}_{i=1}^{N^{\mathrm{tr}}-N^{\mathrm{cri}}})]\Big]^{-1}.$$
$$\nabla^3_{\sigma\phi_\lambda\phi_\lambda} E_{P_{\phi^*}(\theta)}[L(\theta, \{\bar{\mathcal{T}}_i^{\mathrm{cri}}(\lambda_i^{\bar{\zeta}_j})\}_{i=1}^{N^{\mathrm{cri}}}, \{\mathcal{T}_i^{\mathrm{tr}}\}_{i=1}^{N^{\mathrm{tr}}-N^{\mathrm{cri}}})],$$

and similarly we can derive the expression of $\nabla^3_{\phi_\lambda\phi_\lambda\phi_\lambda}\sigma^*$. Therefore, we can see that $||\sigma^*||$, $||\nabla_{\phi_\lambda}\sigma^*||$, and $||\nabla^2_{\phi_\lambda\phi_\lambda}\sigma^*||$ are both lower bounded and upper bounded, and thus there exists positive constants $C_I$ and $\bar{C}_I$ such that $||\nabla_{\phi_\lambda} I(\theta; \{\bar{\mathcal{T}}_i^{\mathrm{cri}}(\lambda_i \sim P_{\phi_\lambda}(\lambda))\}_{i=1}^{N^{\mathrm{cri}}}|\{\mathcal{T}_i^{\mathrm{cri}}\}_{i=1}^{N^{\mathrm{cri}}})|| \le C_I$ and $||\nabla^2_{\phi_\lambda\phi_\lambda} I(\theta; \{\bar{\mathcal{T}}_i^{\mathrm{cri}}(\lambda_i \sim P_{\phi_\lambda}(\lambda))\}_{i=1}^{N^{\mathrm{cri}}}|\{\mathcal{T}_i^{\mathrm{cri}}\}_{i=1}^{N^{\mathrm{cri}}})|| \le \bar{C}_I$. $\qquad\square$

For simplicity, we denote $f(\phi_{\lambda,k}) = I(\theta; \{\bar{\mathcal{T}}_i^{\text{cri}}(\lambda_i \sim P_{\phi_\lambda,k}(\lambda))\}_{i=1}^{N^{\text{cri}}} | \{\mathcal{T}_i^{\text{cri}}\}_{i=1}^{N^{\text{cri}}})$. Claim 1 shows that $f(\phi_{\lambda,k})$ is $\bar{C}_I$-smooth, therefore, we have that

$$f(\phi_{\lambda,k+1}) \geq f(\phi_{\lambda,k}) + \langle \nabla_{\phi_\lambda} f(\phi_{\lambda,k}), \phi_{\lambda,k+1} - \phi_{\lambda,k} \rangle - \frac{\bar{C}_I}{2}||\phi_{\lambda,k+1} - \phi_{\lambda,k}||^2,$$

$$\stackrel{(a)}{=} f(\phi_{\lambda,k}) + \beta||\nabla_{\phi_\lambda} f(\phi_{\lambda,k})||^2 - \frac{\bar{C}_I \beta^2}{2}||\nabla_{\phi_\lambda} f(\phi_{\lambda,k})||^2,$$

$$\stackrel{(b)}{\Rightarrow} \beta||\nabla_{\phi_\lambda} f(\phi_{\lambda,k})||^2 \leq f(\phi_{\lambda,k+1}) - f(\phi_{\lambda,k}) + \frac{\bar{C}_I C_I^2 \beta^2}{2}$$

$$\stackrel{(c)}{\Rightarrow} ||\nabla_{\phi_\lambda} f(\phi_{\lambda,k})||^2 \leq \frac{\bar{C}_I \sqrt{K}}{2}[f(\phi_{\lambda,k+1}) - f(\phi_{\lambda,k})] + \frac{C_I^2}{\sqrt{K}},$$

$$\Rightarrow \frac{1}{K}\sum_{k=0}^{K-1} ||\nabla_{\phi_\lambda} f(\phi_{\lambda,k})||^2 \leq \frac{\bar{C}_I}{2\sqrt{K}}[f(\phi_{\lambda,K}) - f(\phi_{\lambda,0})] + \frac{C_I^2}{\sqrt{K}},$$

where $(a)$ follows the fact that $\phi_{\lambda,k+1} = \phi_{\lambda,k} + \beta \nabla_{\phi_\lambda} f(\phi_{\lambda,k})$, $(b)$ follows the fact that $||\nabla_{\phi_\lambda} f(\phi_\lambda)|| \leq C_I$, and $(c)$ follows the fact that $\beta = \frac{2}{\bar{C}_I \sqrt{K}}$.

## H  PROOF OF THEOREM 2

This section proves Theorem 2 via two steps. Step (i): we prove that $I(\theta; \{\bar{\mathcal{T}}_i^{\text{cri}}(\lambda_i \sim P_{\phi_\lambda,k}(\lambda))\}_{i=1}^{N^{\text{cri}}} | \{\mathcal{T}_i^{\text{cri}}\}_{i=1}^{N^{\text{cri}}})$ is monotonically increasing in Claim 2. Step (ii): we provide that $I(\theta; \{\bar{\mathcal{T}}_i^{\text{cri}}(\lambda_i \sim P_{\phi_\lambda,K}(\lambda))\}_{i=1}^{N^{\text{cri}}} | \{\mathcal{T}_i^{\text{cri}}\}_{i=1}^{N^{\text{cri}}}) > 0$.

**Claim 2.** *If $\beta < \frac{2}{\bar{C}_I}$, the conditional mutual information is monotonically increasing, i.e.,* $I(\theta; \{\bar{\mathcal{T}}_i^{cri}(\lambda_i \sim P_{\phi_\lambda,k+1}(\lambda))\}_{i=1}^{N^{cri}} | \{\mathcal{T}_i^{cri}\}_{i=1}^{N^{cri}}) \geq I(\theta; \{\bar{\mathcal{T}}_i^{cri}(\lambda_i \sim P_{\phi_\lambda,k}(\lambda))\}_{i=1}^{N^{cri}} | \{\mathcal{T}_i^{cri}\}_{i=1}^{N^{cri}})$, *and is strictly increasing if* $||\nabla_{\phi_\lambda} I(\theta; \{\bar{\mathcal{T}}_i^{cri}(\lambda_i \sim P_{\phi_\lambda,k}(\lambda))\}_{i=1}^{N^{cri}} | \{\mathcal{T}_i^{cri}\}_{i=1}^{N^{cri}})|| > 0$.

*Proof.* For simplicity, we denote $f(\phi_{\lambda,k}) = I(\theta; \{\bar{\mathcal{T}}_i^{\text{cri}}(\lambda_i \sim P_{\phi_\lambda,k}(\lambda))\}_{i=1}^{N^{\text{cri}}} | \{\mathcal{T}_i^{\text{cri}}\}_{i=1}^{N^{\text{cri}}})$. Therefore, we have that

$$f(\phi_{\lambda,k+1}) \stackrel{(a)}{\geq} f(\phi_{\lambda,k}) + \langle \nabla_{\phi_\lambda} f(\phi_{\lambda,k}), \phi_{\lambda,k+1} - \phi_{\lambda,k} \rangle - \frac{\bar{C}_I}{2}||\phi_{\lambda,k+1} - \phi_{\lambda,k}||^2,$$

$$\stackrel{(b)}{=} f(\phi_{\lambda,k}) + \beta||\nabla_{\phi_\lambda} f(\phi_{\lambda,k})||^2 - \frac{\bar{C}_I \beta^2}{2}||\nabla_{\phi_\lambda} f(\phi_{\lambda,k})||^2,$$

$$\Rightarrow f(\phi_{\lambda,k+1}) - f(\phi_{\lambda,k}) \geq \frac{2\beta - \bar{C}_I \beta^2}{2}||\nabla_{\phi_\lambda} f(\phi_{\lambda,k})||^2 \geq 0 \qquad (14)$$

where $(a)$ follows the fact that $f(\phi_\lambda)$ is $\bar{C}_I$-smooth (Claim 1), $(b)$ follows the fact that $\phi_{\lambda,k+1} = \phi_{\lambda,k} + \beta \nabla_{\phi_\lambda} f(\phi_{\lambda,k})$. from (14), we can see that $f(\phi_{\lambda,k+1}) \geq f(\phi_{\lambda,k})$. Moreover, $f(\phi_{\lambda,k+1}) > f(\phi_{\lambda,k})$ if $||\nabla_{\phi_\lambda} f(\phi_{\lambda,k})||^2 > 0$. □

From Claim 2, we know that $I(\theta; \{\bar{\mathcal{T}}_i^{\text{cri}}(\lambda_i \sim P_{\phi_\lambda,K}(\lambda))\}_{i=1}^{N^{\text{cri}}} | \{\mathcal{T}_i^{\text{cri}}\}_{i=1}^{N^{\text{cri}}}) \geq I(\theta; \{\bar{\mathcal{T}}_i^{\text{cri}}(\lambda_i \sim P_{\phi_\lambda,0}(\lambda))\}_{i=1}^{N^{\text{cri}}} | \{\mathcal{T}_i^{\text{cri}}\}_{i=1}^{N^{\text{cri}}})$. The only situation where $I(\theta; \{\bar{\mathcal{T}}_i^{\text{cri}}(\lambda_i \sim P_{\phi_\lambda,K}(\lambda))\}_{i=1}^{N^{\text{cri}}} | \{\mathcal{T}_i^{\text{cri}}\}_{i=1}^{N^{\text{cri}}}) = I(\theta; \{\bar{\mathcal{T}}_i^{\text{cri}}(\lambda_i \sim P_{\phi_\lambda,0}(\lambda))\}_{i=1}^{N^{\text{cri}}} | \{\mathcal{T}_i^{\text{cri}}\}_{i=1}^{N^{\text{cri}}})$ is that $\nabla_{\phi_\lambda} I(\theta; \{\bar{\mathcal{T}}_i^{\text{cri}}(\lambda_i \sim P_{\phi_\lambda,0}(\lambda))\}_{i=1}^{N^{\text{cri}}} | \{\mathcal{T}_i^{\text{cri}}\}_{i=1}^{N^{\text{cri}}}) = 0$, i.e., the initialization is a stationary point, which is of zero probability. Therefore, we know that $I(\theta; \{\bar{\mathcal{T}}_i^{\text{cri}}(\lambda_i \sim P_{\phi_\lambda,K}(\lambda))\}_{i=1}^{N^{\text{cri}}} | \{\mathcal{T}_i^{\text{cri}}\}_{i=1}^{N^{\text{cri}}}) > I(\theta; \{\bar{\mathcal{T}}_i^{\text{cri}}(\lambda_i \sim P_{\phi_\lambda,0}(\lambda))\}_{i=1}^{N^{\text{cri}}} | \{\mathcal{T}_i^{\text{cri}}\}_{i=1}^{N^{\text{cri}}})$. Since conditional mutual information is always nonnegative (Wyner, 1978), we know that $I(\theta; \{\bar{\mathcal{T}}_i^{\text{cri}}(\lambda_i \sim P_{\phi_\lambda,K}(\lambda))\}_{i=1}^{N^{\text{cri}}} | \{\mathcal{T}_i^{\text{cri}}\}_{i=1}^{N^{\text{cri}}}) > I(\theta; \{\bar{\mathcal{T}}_i^{\text{cri}}(\lambda_i \sim P_{\phi_\lambda,0}(\lambda))\}_{i=1}^{N^{\text{cri}}} | \{\mathcal{T}_i^{\text{cri}}\}_{i=1}^{N^{\text{cri}}}) \geq 0$.

## I  PROOF OF LEMMA 2

In this section, we prove that the learned augmentation $P_{\phi_\lambda,K}(\lambda)$ imposes a quadratic regularization on the original meta-objective. Let's first consider $\bar{J}_i^{\text{cri}}(\pi_i^{\text{cri}}(\theta), \lambda_i)$. We use $\phi_i$ to denote the

parameter of the task-specific adaptation, i.e., $\pi_{\phi_i} = \pi_i^{\mathrm{cri}}(\theta)$. Since we use MAML to compute the task-specific adaptation, we know that $\phi_i = \theta - \alpha \nabla_\theta J_i^{\mathrm{cri}}(\pi_\theta)$. We use $\bar{s}_{jj'}(\lambda)$ and $\bar{a}_{jj'}(\lambda)$ to represent $\bar{s}_{jj'}$ and $\bar{a}_{jj'}$ to highlight the mixture coefficient $\lambda$. Therefore, we have that

$$E_{\lambda_i \sim \mathcal{N}(\mu_{\lambda,K}, \sigma^2_{\lambda,K})} \Big[ \bar{J}_i^{\mathrm{cri}}(\pi_i^{\mathrm{cri}}(\theta), \lambda_i) \Big],$$

$$= E_{(s_j,a_j),(s'_j,a'_j) \sim \rho^{\pi_{\phi_i}}, \lambda_i \sim \mathcal{N}(\mu_{\lambda,K}, \sigma^2_{\lambda,K})} \Big[ \log \pi_{\phi_i}(\bar{a}_{jj'}(\lambda_i)|\bar{s}_{jj'}(\lambda_i))[\lambda_i A_i^{\pi_{\phi_i}}(s_j, a_j)$$

$$+ (1-\lambda_i) A_i^{\pi_{\phi_i}}(s'_j, a'_j)] \Big],$$

$$\overset{(a)}{=} E_{(s_j,a_j),(s'_j,a'_j) \sim \rho^{\pi_{\phi_i}}, \lambda_i \sim \mathcal{N}(\mu_{\lambda,K}, \sigma^2_{\lambda,K})} \Big[ \log \pi_{\phi_i}(\bar{a}_{jj'}(\lambda_i)|\bar{s}_{jj'}(\lambda_i))\lambda_i A_i^{\pi_{\phi_i}}(s_j, a_j) \Big]$$

$$+ E_{(s_j,a_j),(s'_j,a'_j) \sim \rho^{\pi_{\phi_i}}, \lambda_i \sim \mathcal{N}(\mu_{\lambda,K}, \sigma^2_{\lambda,K})} \Big[ \log \pi_{\phi_i}(\bar{a}_{j'j}(1-\lambda_i)|\bar{s}_{j'j}(1-\lambda_i))(1-\lambda_i) A_i^{\pi_{\phi_i}}(s'_j, a'_j) \Big],$$

$$\overset{(b)}{=} E_{(s_j,a_j),(s'_j,a'_j) \sim \rho^{\pi_{\phi_i}}, \lambda_i \sim \mathcal{N}(\mu_{\lambda,K}, \sigma^2_{\lambda,K})} \Big[ \log \pi_{\phi_i}(\bar{a}_{jj'}(\lambda_i)|\bar{s}_{jj'}(\lambda_i))\lambda_i A_i^{\pi_{\phi_i}}(s_j, a_j) \Big]$$

$$+ E_{(s_j,a_j),(s'_j,a'_j) \sim \rho^{\pi_{\phi_i}}, \lambda_i \sim \mathcal{N}(1-\mu_{\lambda,K}, \sigma^2_{\lambda,K})} \Big[ \log \pi_{\phi_i}(\bar{a}_{jj'}(\lambda_i)|\bar{s}_{jj'}(\lambda_i))\lambda_i A_i^{\pi_{\phi_i}}(s_j, a_j) \Big],$$

$$= E_{(s_j,a_j),(s'_j,a'_j) \sim \rho^{\pi_{\phi_i}}, \lambda_i \sim \mathcal{N}(1, 2\sigma^2_{\lambda,K})} \Big[ \log \pi_{\phi_i}(\bar{a}_{jj'}(\lambda_i)|\bar{s}_{jj'}(\lambda_i))\lambda_i A_i^{\pi_{\phi_i}}(s_j, a_j) \Big], \tag{15}$$

where $(a)$ follows the fact that $s_{jj'}(\lambda) = s_{j'j}(1-\lambda)$ and $a_{jj'}(\lambda) = a_{j'j}(1-\lambda)$, $(b)$ follows the fact that $(1-\lambda_i) \sim \mathcal{N}(1-\mu_{\lambda,K}, \sigma^2_{\lambda,K})$ if $\lambda_i \sim \mathcal{N}(\mu_{\lambda,K}, \sigma^2_{\lambda,K})$. Let $x_i = 1 - \lambda_i$ and $F_i(x_i) = \log \pi_{\phi_i}(\bar{a}_{jj'}(\lambda_i)|\bar{s}_{jj'}(\lambda_i))\lambda_i A_i^{\pi_{\phi_i}}(s_j, a_j)$, therefore, the second-order approximation of $F_i(x_i)$ is

$$F_i(x_i) \approx F_i(0) + F_i'(0)x_i + \frac{1}{2}F_i''(0)x_i^2. \tag{16}$$

We now derive the expression of $F_i'(0)$ and $F_i''(0)$.

$$F_i'(x_i) = \frac{\partial F_i(x_i)}{\partial \bar{a}_{jj'}(\lambda)}\frac{\partial \bar{a}_{jj'}(\lambda)}{\partial x_i} + \frac{\partial F_i(x_i)}{\partial \bar{s}_{jj'}(\lambda)}\frac{\partial \bar{s}_{jj'}(\lambda)}{\partial x_i} + \frac{\partial F_i(x_i)}{\partial x_i},$$

$$\overset{(c)}{=} \frac{\partial F_i(x_i)}{\partial \bar{s}_{jj'}(\lambda)}\frac{\partial \bar{s}_{jj'}(\lambda)}{\partial x_i} + \frac{\partial F_i(x_i)}{\partial x_i},$$

$$= \lambda_i A_i^{\pi_{\phi_i}}(s_j, a_j)(\nabla_s \log \pi_{\phi_i}(\bar{a}_{jj'}(\lambda_i)|\bar{s}_{jj'}(\lambda_i)))^\top (s_{j'} - s_j)$$

$$- \log \pi_{\phi_i}(\bar{a}_{jj'}(\lambda_i)|\bar{s}_{jj'}(\lambda_i))A_i^{\pi_{\phi_i}}(s_j, a_j),$$

$$\Rightarrow F_i'(0) = A_i^{\pi_{\phi_i}}(s_j, a_j)(\nabla_s \log \pi_{\phi_i}(a_j|s_j))^\top (s_{j'} - s_j) - \log \pi_{\phi_i}(a_j|s_j)A_i^{\pi_{\phi_i}}(s_j, a_j), \tag{17}$$

where $(c)$ follows the fact that $\frac{\partial \bar{a}_{jj'}(\lambda)}{\partial x_i} = 0$ almost everywhere. We now reason about the second-order derivation:

$$F_i''(x_i) = \frac{\partial \lambda_i A_i^{\pi_{\phi_i}}(s_j, a_j)(\nabla_s \log \pi_{\phi_i}(\bar{a}_{jj'}(\lambda_i)|\bar{s}_{jj'}(\lambda_i)))^\top (s_{j'} - s_j)}{\partial x_i}$$

$$- \frac{\partial \log \pi_{\phi_i}(\bar{a}_{jj'}(\lambda_i)|\bar{s}_{jj'}(\lambda_i))A_i^{\pi_{\phi_i}}(s_j, a_j)}{\partial x_i},$$

$$= -A_i^{\pi_{\phi_i}}(s_j, a_j)(\nabla_s \log \pi_{\phi_i}(\bar{a}_{jj'}(\lambda_i)|\bar{s}_{jj'}(\lambda_i)))^\top (s_{j'} - s_j)$$

$$= \lambda_i A_i^{\pi_{\phi_i}}(s_j, a_j)(s_{j'} - s_j)^\top (\nabla^2_{ss} \log \pi_{\phi_i}(\bar{a}_{jj'}(\lambda_i)|\bar{s}_{jj'}(\lambda_i)))(s_{j'} - s_j)$$

$$- A_i^{\pi_{\phi_i}}(s_j, a_j)(\nabla_s \log \pi_{\phi_i}(\bar{a}_{jj'}(\lambda_i)|\bar{s}_{jj'}(\lambda_i)))^\top (s_{j'} - s_j),$$

$$\Rightarrow F_i''(0) = -2A_i^{\pi_{\phi_i}}(s_j, a_j)(\nabla_s \log \pi_{\phi_i}(a_j|s_j))^\top (s_{j'} - s_j)$$

$$+ A_i^{\pi_{\phi_i}}(s_j, a_j)(s_{j'} - s_j)^\top (\nabla^2_{ss} \log \pi_{\phi_i}(a_j|s_j))(s_{j'} - s_j). \tag{18}$$

By plugging (17)-(18) into (16), we have that

$$F_i(x_i) \approx \log \pi_{\phi_i}(a_j|s_j)A_i^{\pi_{\phi_i}}(s_j, a_j)$$

$$+ \Big[ A_i^{\pi_{\phi_i}}(s_j, a_j)(\nabla_s \log \pi_{\phi_i}(a_j|s_j))^\top (s_{j'} - s_j) - \log \pi_{\phi_i}(a_j|s_j)A_i^{\pi_{\phi_i}}(s_j, a_j) \Big]x_i$$

$$- 2A_i^{\pi_{\phi_i}}(s_j, a_j)(\nabla_s \log \pi_{\phi_i}(a_j|s_j))^\top (s_{j'} - s_j)x_i^2$$
$$+ A_i^{\pi_{\phi_i}}(s_j, a_j)(s_{j'} - s_j)^\top (\nabla_{ss}^2 \log \pi_{\phi_i}(a_j|s_j))(s_{j'} - s_j)x_i^2,$$
$$= \log \pi_{\phi_i}(a_j|s_j)A_i^{\pi_{\phi_i}}(s_j, a_j) + C_{\lambda_i}(s_j, a_j)$$
$$+ A_i^{\pi_{\phi_i}}(s_j, a_j)(s_{j'} - s_j)^\top (\nabla_{ss}^2 \log \pi_{\phi_i}(a_j|s_j))(s_{j'} - s_j)x_i^2, \tag{19}$$

where $C_{\lambda_i}(s_j, a_j) = \Big[ A_i^{\pi_{\phi_i}}(s_j, a_j)(\nabla_s \log \pi_{\phi_i}(a_j|s_j))^\top (s_{j'} - s_j) - \log \pi_{\phi_i}(a_j|s_j)A_i^{\pi_{\phi_i}}(s_j, a_j) \Big](1 - \lambda_i) - 2A_i^{\pi_{\phi_i}}(s_j, a_j)(\nabla_s \log \pi_{\phi_i}(a_j|s_j))^\top (s_{j'} - s_j)(1 - \lambda_i)^2.$

Now we take a look at the term $\nabla_{ss}^2 \log \pi_{\phi_i}(a_j|s_j)$. Recall that the softmax policy parameterization $\pi_{\phi_i}(a|s) = \frac{e^{\phi_i^\top f(s,a)}}{\sum_{a' \in \mathcal{A}} e^{\phi_i^\top f(s,a')}}$, therefore we have that

$$\nabla_{ss}^2 \log \pi_{\phi_i}(a|s) = \nabla_{ss}^2 \Big[ \phi_i^\top f(s,a) - \log \sum_{a' \in \mathcal{A}} e^{\phi_i^\top f(s,a')} \Big],$$

$$= \phi_i^\top \nabla_{ss}^2 f(s,a) - \frac{\sum_{a' \in \mathcal{A}} \phi_i^\top \nabla_{ss}^2 f(s,a')e^{\phi_i^\top f(s,a')} + \phi_i^\top (\nabla_s f(s,a'))(\nabla_s f(s,a'))^\top e^{\phi_i^\top f(s,a')}\phi_i}{\sum_{a' \in \mathcal{A}} e^{\phi_i^\top f(s,a')}}$$

$$+ \frac{(\sum_{a' \in \mathcal{A}} \phi_i^\top \nabla_s f(s,a')e^{\phi_i^\top f(s,a')})^2}{(\sum_{a' \in \mathcal{A}} e^{\phi_i^\top f(s,a')})^2},$$

$$= \phi_i^\top \nabla_{ss}^2 f(s,a) - \frac{\sum_{a' \in \mathcal{A}} \phi_i^\top \nabla_{ss}^2 f(s,a')e^{\phi_i^\top f(s,a')}}{\sum_{a' \in \mathcal{A}} e^{\phi_i^\top f(s,a')}}$$

$$- \phi_i^\top \Big[ \frac{[\sum_{a' \in \mathcal{A}}(\nabla_s f(s,a'))(\nabla_s f(s,a'))^\top e^{\phi_i^\top f(s,a')}](\sum_{a' \in \mathcal{A}} e^{\phi_i^\top f(s,a')}) - (\sum_{a' \in \mathcal{A}} \nabla_s f(s,a')e^{\phi_i^\top f(s,a')})^2}{(\sum_{a' \in \mathcal{A}} e^{\phi_i^\top f(s,a')})^2} \Big]\phi_i,$$

$$= \phi_i^\top \nabla_{ss}^2 f(s,a) - \frac{\sum_{a' \in \mathcal{A}} \phi_i^\top \nabla_{ss}^2 f(s,a')e^{\phi_i^\top f(s,a')}}{\sum_{a' \in \mathcal{A}} e^{\phi_i^\top f(s,a')}} - \phi_i^\top H(s,a)\phi_i, \tag{20}$$

where $H(s,a) = \frac{[\sum_{a' \in \mathcal{A}}(\nabla_s f(s,a'))(\nabla_s f(s,a'))^\top e^{\phi_i^\top f(s,a')}](\sum_{a' \in \mathcal{A}} e^{\phi_i^\top f(s,a')}) - (\sum_{a' \in \mathcal{A}} \nabla_s f(s,a')e^{\phi_i^\top f(s,a')})^2}{(\sum_{a' \in \mathcal{A}} e^{\phi_i^\top f(s,a')})^2} \succ 0$ by Cauchy-Schwartz inequality. By plugging (20) into (19), we have that

$$F_i(x_i) \approx \log \pi_{\phi_i}(a_j|s_j)A_i^{\pi_{\phi_i}}(s_j, a_j) + C_{\lambda_i}(s_j, a_j) + A_i^{\pi_{\phi_i}}(s_j, a_j)(s_{j'} - s_j)^\top \cdot$$
$$\Big[ \phi_i^\top \nabla_{ss}^2 f(s,a) - \frac{\sum_{a' \in \mathcal{A}} \phi_i^\top \nabla_{ss}^2 f(s,a')e^{\phi_i^\top f(s,a')}}{\sum_{a' \in \mathcal{A}} e^{\phi_i^\top f(s,a')}} - \phi_i^\top H(s_j, a_j)\phi_i \Big](s_{j'} - s_j)x_i^2,$$

$$= \log \pi_{\phi_i}(a_j|s_j)A_i^{\pi_{\phi_i}}(s_j, a_j) + \bar{C}_{\lambda_i}(s_j, a_j) - \phi_i^\top \bar{H}_{\lambda_i}^{\mathrm{cri}}(s_j, a_j)\phi_i,$$

$$\overset{(d)}{=} \log \pi_{\phi_i}(a_j|s_j)A_i^{\pi_{\phi_i}}(s_j, a_j) + \bar{C}_{\lambda_i}(s_j, a_j) - (\theta - \alpha\nabla_\theta J_i^{\mathrm{cri}}(\pi_\theta))^\top \bar{H}_{\lambda_i}^{\mathrm{cri}}(s_j, a_j)(\theta - \alpha\nabla_\theta J_i^{\mathrm{cri}}(\pi_\theta)),$$

$$= \log \pi_{\phi_i}(a_j|s_j)A_i^{\pi_{\phi_i}}(s_j, a_j) + \tilde{C}_{\lambda_i}(s_j, a_j) - \theta^\top \bar{H}_{\lambda_i}^{\mathrm{cri}}(s_j, a_j)\theta, \tag{21}$$

where $(d)$ follows the fact that $\phi_i = \theta - \alpha\nabla_\theta J_i^{\mathrm{cri}}(\pi_\theta)$, $\bar{C}_{\lambda_i}(s_j, a_j) = A_i^{\pi_{\phi_i}}(s_j, a_j)(s_{j'} - s_j)^\top \Big[ \phi_i^\top \nabla_{ss}^2 f(s,a) - \frac{\sum_{a' \in \mathcal{A}} \phi_i^\top \nabla_{ss}^2 f(s,a')e^{\phi_i^\top f(s,a')}}{\sum_{a' \in \mathcal{A}} e^{\phi_i^\top f(s,a')}} \Big](s_{j'} - s_j)x_i^2$, $\bar{H}_{\lambda_i}^{\mathrm{cri}}(s_j, a_j) = A_i^{\pi_{\phi_i}}(s_j, a_j)H(s_j, a_j)(s_{j'} - s_j)x_i^2 \succ 0$ given that $H(s_j, a_j) \succ 0$, and $\tilde{C}_{\lambda_i}(s,a) = \bar{C}_{\lambda_i}(s,a) - \alpha^2(\nabla_\theta J_i^{\mathrm{cri}}(\pi_\theta))^\top \bar{H}_{\lambda_i}^{\mathrm{cri}}(s,a)(\nabla_\theta J_i^{\mathrm{cri}}(\pi_\theta))$.

Therefore, we have that

$$\bar{J}_i^{\mathrm{cri}}(\pi_i^{\mathrm{cri}}(\theta), \lambda_i) = E_{(s_j, a_j), (s_{j'}, a_{j'}) \sim \rho^{\pi_{\phi_i}}}[F_i(x_i)]$$

$$\overset{(e)}{=} E_{(s_j, a_j) \sim \rho^{\pi_{\phi_i}}}\Big[ \log \pi_{\phi_i}(a_j|s_j)A_i^{\pi_{\phi_i}}(s_j, a_j) + \tilde{C}_{\lambda_i}(s_j, a_j) - \theta^\top \bar{H}_{\lambda_i}^{\mathrm{cri}}(s_j, a_j)\theta \Big],$$

$$= J_i^{\mathrm{cri}}(\pi_i^{\mathrm{cri}}(\theta)) + \tilde{C}_{\lambda_i} - \theta^\top \bar{H}_{\lambda_i}^{\mathrm{cri}}\theta,$$

where $(e)$ follows (21), $\tilde{C}_{\lambda_i} = E_{(s_j,a_j)\sim\rho^{\pi_{\phi_i}}}[\tilde{C}_{\lambda_i}(s_j,a_j)]$, and $\bar{H}_{\lambda_i}^{\mathrm{cri}} = E_{(s_j,a_j)\sim\rho^{\pi_{\phi_i}}}[\bar{H}_{\lambda_i}^{\mathrm{cri}}(s_j,a_j)] \succ 0$ given that $\bar{H}_{\lambda_i}^{\mathrm{cri}}(s_j,a_j) \succ 0$. If we only consider the second-order term, we can see that $\bar{J}_i^{\mathrm{cri}}(\pi_i^{\mathrm{cri}}(\theta),\lambda_i) \approx J_i^{\mathrm{cri}}(\pi_i^{\mathrm{cri}}(\theta)) - \theta^\top \bar{H}_{\lambda_i}^{\mathrm{cri}}\theta$. Therefore, we have that $L(\theta, \{\bar{\mathcal{T}}_i^{\mathrm{tr}}(\lambda_i)\}_{i=1}^{N^{\mathrm{cri}}}, \{\mathcal{T}_i^{\mathrm{tr}}\}_{i=1}^{N^{\mathrm{tr}}-N^{\mathrm{cri}}}) \approx L(\theta, \{\mathcal{T}_i^{\mathrm{tr}}\}_{i=1}^{N^{\mathrm{cri}}}) - \theta^\top(\sum_{i=1}^{N^{\mathrm{cri}}}\bar{H}_i^{\mathrm{cri}})\theta$ where $(\sum_{i=1}^{N^{\mathrm{cri}}}\bar{H}_i^{\mathrm{cri}}) \succ 0$ given that $\bar{H}_{\lambda_i}^{\mathrm{cri}} \succ 0$. Thus we have that $E_{\lambda_i\sim P_{\phi_{\lambda,K}}(\lambda)}[L(\theta, \{\bar{\mathcal{T}}_i^{\mathrm{cri}}(\lambda_i)\}_{i=1}^{N^{\mathrm{cri}}}, \{\mathcal{T}_i^{\mathrm{tr}}\}_{i=1}^{N^{\mathrm{tr}}-N^{\mathrm{cri}}})] \approx L(\theta, \{\mathcal{T}_i^{\mathrm{tr}}\}_{i=1}^{N^{\mathrm{tr}}}) - \theta^\top(\sum_{i=1}^{N^{\mathrm{cri}}}\bar{H}_i^{\mathrm{cri}})\theta$ where $\bar{H}_i^{\mathrm{cri}} = E_{\lambda_i\sim P_{\phi_{\lambda,K}}(\lambda)}[\bar{H}_{\lambda_i}^{\mathrm{cri}}] \succ 0$ given that $\bar{H}_{\lambda_i}^{\mathrm{cri}} \succ 0$.

## J  PROOF OF THEOREM 3

We start with standard uniform deviation bound based on Rademacher complexity (Bartlett & Mendelson, 2002).

**Claim 3** ((Bartlett & Mendelson, 2002)). *Let the sample $\{z_1,\cdots,z_N\}$ be drawn i.i.d. from a distribution $P$ over $\mathcal{Z}$ and let $\mathcal{F}$ be a function class on $\mathcal{Z}$ mapping from $\mathcal{Z}$ to a bounded set. Then for $\delta > 0$, with probability at least $1-\delta$, it holds that $\sup_{f\sim\mathcal{F}}||\frac{1}{N}\sum_{i=1}^N f(z_i) - E_{z\sim P}[f(z)]|| \le 2R(\mathcal{F},z_1,\cdots,z_n) + \sqrt{\frac{\log(1/\delta)}{N}}$, where $R(\mathcal{F},z_i,\cdots,z_N)$ is the Rademacher complexity of the function class $\mathcal{F}$.*

From Claim 3, we know that the generalization gap $|\mathcal{G}(\mathcal{F}_\gamma)| \le R(\bar{\mathcal{F}}_{\bar\gamma},\mathcal{T}_1^{\mathrm{tr}},\cdots,\mathcal{T}_{N^{\mathrm{tr}}}^{\mathrm{tr}}) + \sqrt{\frac{\log(1/\delta)}{N^{\mathrm{tr}}}}$, where $\bar{F}_{\bar\gamma} \triangleq \{J_i(\pi_\theta) : \pi_\theta \in \mathcal{F}_{\bar\gamma}\}$. Therefore, we can compute the Rademacher complexity:

$$
\begin{aligned}
R(\bar{\mathcal{F}}_{\bar\gamma},\mathcal{T}_1^{\mathrm{tr}},\cdots,\mathcal{T}_{N^{\mathrm{tr}}}^{\mathrm{tr}}) &= E_{\sigma_i}\Big[\sup_{J\sim\bar{F}_{\bar\gamma}}\frac{1}{N^{\mathrm{tr}}}\sum_{i=1}^{N^{\mathrm{tr}}}\sigma_i J_i^{\mathrm{tr}}(\pi_i(\theta))\Big], \\
&\le \sup_{\pi_\theta\sim\mathcal{F}_{\bar\gamma},i\sim P(\mathcal{T})} J_i(\pi_i(\theta)), \\
&= \sup_{\pi_\theta\sim\mathcal{F}_{\bar\gamma},i\sim P(\mathcal{T})} E_{(s,a)\sim\rho^{\pi_{\phi_i}}}^{\pi_{\phi_i}}[\log\pi_{\phi_i}(a|s)A_i^{\pi_{\phi_i}}(s,a)], \\
&= \sup_{\pi_\theta\sim\mathcal{F}_{\bar\gamma},i\sim P(\mathcal{T})} E_{(s,a)\sim\rho^{\pi_{\phi_i}}}^{\pi_{\phi_i}}[(\phi_i^\top f(s,a) - \log(\sum_{a'\in\mathcal{A}}e^{\phi_i^\top f(s,a)}))A_i^{\pi_{\phi_i}}(s,a)],
\end{aligned}
$$

where $\sigma_i$ is a random variable with equal probability of choose 1 and $-1$. Recall that $\phi_i = \theta - \alpha\nabla_\theta J_i(\pi_\theta)$ and $||\nabla_\theta J_i(\pi_\theta)||$ is bounded. Moreover, $A_i^{\pi_{\phi_i}}(s,a)$ is also bounded given that the reward value is bounded, and the chosen feature vector $f(s,a)$ is also bounded. Therefore, there exists a constant $C_1$ such that $R(\bar{\mathcal{F}}_{\bar\gamma},\mathcal{T}_1^{\mathrm{tr}},\cdots,\mathcal{T}_{N^{\mathrm{tr}}}^{\mathrm{tr}}) \le \frac{C_1}{\sqrt{N^{\mathrm{tr}}}}\sup_{\pi_\theta\sim\mathcal{F}_{\bar\gamma},i\sim P(\mathcal{T})} E_{(s,a)\sim\rho^{\pi_{\phi_i}}}^{\pi_{\phi_i}}[\theta^\top\bar{h}_i]$ where $\bar{h}_i^\top\bar{h}_i = E_{i\sim P(\mathcal{T})}[\bar{H}_i]$. Therefore, we have that $R(\bar{\mathcal{F}}_{\bar\gamma},\mathcal{T}_1^{\mathrm{tr}},\cdots,\mathcal{T}_{N^{\mathrm{tr}}}^{\mathrm{tr}}) \le C_2\sqrt{\frac{\gamma}{N^{\mathrm{tr}}}}$ where $C_2$ is a positive constant. Therefore, we have that $|\mathcal{G}(\mathcal{F}_\gamma)| \le 2C_2\sqrt{\frac{\gamma}{N^{\mathrm{tr}}}} + \sqrt{\frac{\log(1/\delta)}{N^{\mathrm{tr}}}} = O(\sqrt{\frac{\gamma}{N^{\mathrm{tr}}}} + \sqrt{\frac{\log(1/\delta)}{N^{\mathrm{tr}}}})$.

## K  EXPERIMENT DETAILS

### K.1  DRONE NAVIGATION WITH OBSTACLES

We cannot directly train the meta-learning algorithm on the physical drone because during training, the drone needs to interact with the environment and can be damaged due to collision with the obstacle and the wall. To avoid the damage of the drone, we build a simulator in Gazebo (Figure 2) Liu & Zhu (2022; 2024a) that imitates the physical environment with the scale $1:1$. We train the meta-learning algorithm on the simulated drone in the simulator and the empirical results (i.e., successful rate) are counted in the simulator. Once we obtain a learned policy that has good performance in the simulator, we implement the policy on the physical drone.

**Discussion of the sim-to-real problem.** In some cases, the models that have good performance in the simulator may not have good performance in the real world due to the reason that the simulator cannot $100\%$ precisely imitate the physical world. However, in our case, the sim-to-real issue is not

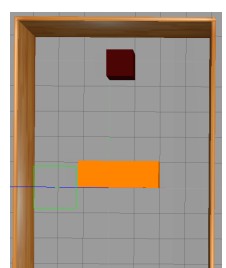

significant because of two reasons: (i) the simulated drone is built according to the dynamics of a real Ar. Drone 2.0 (Huang & Sturm, 2014); (ii) the states and actions are just the coordinates of the location and the heading direction of the drone instead of some low-level control such as the motor's velocity, etc. Given that Vicon can output precise pose of the physical drone and the simulator is built on the $1 : 1$ scale. If a learned trajectory can succeed in the simulator, it can succeed in the real world given that the low-level control of both the simulated and physical drones are given.

In this experiment, the state of the drone is its 3-D coordinate $(x, y, z)$ and the action of the drone is also a 3-D coordinate $(dx, dy, dz)$ which captures the heading direction of the drone. We fix the length of each step as $0.1$ and thus the next state is $(x + \frac{dx}{10\sqrt{(dx)^2+(dy)^2+(dz)^2}}, y + \frac{dy}{10\sqrt{(dx)^2+(dy)^2+(dz)^2}}, z + \frac{dz}{10\sqrt{(dx)^2+(dy)^2+(dz)^2}})$. In this experiment, we do not need the drone to change its height so that we usually fix the value of $z$ and set $dz = 0$. The goal is an $1 \times 1$ square. Denote the coordinate of the center of the goal as $(x_{\text{goal}}, y_{\text{goal}})$, then for all the different tasks, $x_{\text{goal}} \in (0.5, 6.5)$ and $y_{\text{goal}} \in (10, 11)$. The obstacle is a $3 \times 1$ square. Denote the coordinate of the lower left end of the obstacle as $(x_{\text{obstacle}}, y_{\text{obstacle}})$, the for the different tasks, $x_{\text{obstacle}} \in (0, 4)$ and $y_{\text{obstacle}} \in (4, 5)$.

we first use the 50 training tasks to learn a meta-policy. We then randomly sample 10 validation tasks and find the top 5 validation tasks where the meta-policy adapts with the worst performance. These 5 tasks are the poorly-adapted tasks. Note that these 5 poorly-adapted tasks are not included in the 20 test tasks when we evaluate the generalization of our algorithm. We find 5 critical tasks from the 20 training tasks.

### K.2 STOCK MARKET

We use the real-world data of 30 constitute stocks in Dow Jones Industrial Average from 2021-01-01 to 2022-01-01. The 30 stocks are respectively: 'AXP', 'AMGN', 'AAPL', 'BA', 'CAT', 'CSCO', 'CVX', 'GS', 'HD', 'HON', 'IBM', 'INTC', 'JNJ', 'KO', 'JPM', 'MCD', 'MMM', 'MRK', 'MSFT', 'NKE', 'PG', 'TRV', 'UNH', 'CRM', 'VZ', 'V', 'WBA', 'WMT', 'DIS', 'DOW'.

The state of the stock market MDP is the perception of the stock market, including the open/close price of each stock, the current asset, and some technical indices (Liu et al., 2021). The action has the same dimension as the number of stocks where each dimension represents the amount of buying/selling the corresponding stock. The detailed formulation of the MDP can be found in FinRL (Liu et al., 2021).

The turbulence index is a technical index of stock market and is included as a dimension of the state (Liu et al., 2021). The turbulence index measures the price fluctuation of a stock. If the turbulence index is high, the corresponding stock has a high fluctuating price and thus is risky to buy. Therefore, an investor unwilling to take risks has a relatively low turbulence threshold. The function $p_2$ is defined as the amount of buying the stocks whose turbulence index is larger than the turbulence threshold. Therefore, the more the target investor buys the stocks whose turbulence index is larger than the turbulence threshold, the larger $p_2$ will be and thus the smaller reward the target investor will receive. We choose the turbulence threshold between $45$ and $50$.

We randomly sample 10 validation tasks and find the top 5 validation tasks where the meta-policy adapts with the worst performance. These 5 tasks are the poorly-adapted tasks. We find 5 critical tasks from the 20 training tasks.

### K.3 MUJOCO

The target velocity of all the three robots (i.e., Halfcheetah, Hopper, and Walker2d) is between $0$ and $2$. Note that we fix the training tasks and we first use these $50$ training tasks to learn a meta-policy. We then randomly sample 10 validation tasks and find the top 5 validation tasks where the meta-policy adapts with the worst performance. These 5 tasks are the poorly-adapted tasks. Note that these 5 poorly-adapted tasks are not included in the 20 test tasks when we evaluate the generalization of our algorithm. We find 5 critical tasks from the 20 training tasks.

## K.4 EVALUATION OF THE EXPLANATION

This section evaluates the fidelity and usefulness of the explanation.

**Evaluation of fidelity**. Fidelity means the correctness of the explanation. Recall that the explanation (i.e., the critical tasks) aims to identify the most important training tasks to achieve high cumulative reward on the poorly-adapted tasks. To evaluate the fidelity, we train a meta-policy on the critical tasks and evaluate the performance of the meta-policy on the poorly adapted tasks. We introduce two baselines for comparison. The first baseline is the "original meta-policy" that trains on all the training tasks. We refer to this baseline as "original". The second baseline is that we randomly pick $N^{\text{cri}} = 5$ training tasks and train a meta-policy over the $N^{\text{cri}} = 5$ training tasks. We refer to this baseline as "random". We compare the performance on the poorly-adapted tasks with these two baselines.

Table 2: Fidelity comparison

|  | Drone | Stock market | HalfCheetah | Hopper | Walker |
|---|---|---|---|---|---|
| Ours | $0.97 \pm 0.02$ | $442.29 \pm 12.79$ | $-50.16 \pm 3.32$ | $-7.71 \pm 2.43$ | $-49.26 \pm 4.27$ |
| Original | $0.68 \pm 0.16$ | $296.27 \pm 35.16$ | $-104.79 \pm 12.72$ | $-46.27 \pm 8.62$ | $-108.38 \pm 12.29$ |
| Random | $0.71 \pm 0.08$ | $284.97 \pm 29.85$ | $-96.78 \pm 9.24$ | $-52.91 \pm 6.36$ | $-95.27 \pm 17.46$ |

Table 2 shows that our explanation has high fidelity because the meta-policy trained on our explanation significantly outperforms the two baselines on the poorly-adapted tasks.

**Evaluation of usefulness**. Usefulness means whether the explanation can indeed help improve generalization. Table 1 already shows that our method (XMRL-G) can significantly improve MAML. However, this might be the effect of the task augmentation method. To evaluate whether the critical tasks help improve generalization. We randomly pick $N^{\text{cri}} = 5$ training tasks and use the same algorithm (Algorithm 1) to augment these 5 tasks. We refer to this method as random, and we compare the generalization of our method with this random method.

Table 3: Usefulness comparison

|  | Drone | Stock market | HalfCheetah | Hopper | Walker |
|---|---|---|---|---|---|
| MAML | $0.87 \pm 0.01$ | $359.13 \pm 18.63$ | $-68.89 \pm 4.36$ | $-23.24 \pm 5.71$ | $-82.18 \pm 6.64$ |
| Ours | $0.96 \pm 0.02$ | $426.36 \pm 17.15$ | $-53.88 \pm 5.21$ | $-12.50 \pm 2.37$ | $-55.76 \pm 5.01$ |
| Random | $0.88 \pm 0.02$ | $371.24 \pm 17.81$ | $-66.81 \pm 6.65$ | $-22.69 \pm 4.60$ | $-78.44 \pm 9.33$ |

Table 3 shows that our explanation has high usefulness because randomly pick $N^{\text{cri}} = 5$ training tasks and augment can only slightly improve the generalization, while our method can significantly improve generalization.

