# OpenReview forum: "Improving Generalization of Meta Reinforcement Learning via Explanation"
_ICLR.cc/2025/Conference — Submitted to ICLR 2025_

### Official Review · Reviewer_pVJk · 2024-10-17

**Soundness:** 1
**Presentation:** 2
**Contribution:** 2
**Rating:** 3
**Confidence:** 3

**Summary:**

Focusing on the imbalanced generalization issue, the authors propose to identify the critical tasks among all the training tasks. After extracting the critical tasks, the authors propose to optimally augment the critical tasks and then achieve overall generalization performance improvement.
With thorough theoretical justification, the authors provide the convergence rate and the generalization performance gaps.
Finally, the authors demonstrate evaluation results in both real-world and simulation environments.

**Strengths:**

1. The paper gives a feasible explanation of the generalization problem w.r.t meta RL setting, namely treating all the training tasks equally.
2. The paper provides a thorough theoretical analysis to validate the proposed algorithm.
3. The evaluation results are even conducted in the real world.

**Weaknesses:**

1. My biggest concern is centered around the experiment. Except for the evaluation results on the real world (which I appreciated before) and MuJoCo, there are no evaluation results on the algorithmic design and its effectiveness. It would be more appreciated if an ablation study could be added to show the effectiveness of the optimal augmenting strategy w.r.t critical tasks. Furthermore, in my view, poor tasks could have a large impact on the design of the whole algorithm, that says, if we set the poor tasks as the whole validation tasks, it seems that using the explanation (importance vector) is enough, why should choose to augment the tasks. Hence, it is necessary to point out the significance of augmenting the tasks.
2. The paper states that the performance of non-critical tasks could not be affected even if augmenting the critical tasks and theoretically proves this claim based on conditional mutual information. However, I am skeptical about using the difference in conditional mutual information equal to 0 to prove the variation of performance. If the quantity of the augmented critical tasks is much larger than the non-critical tasks, the algorithm would overfit these augmented critical tasks and ignore the non-critical tasks. Hence, the performance on non-critical tasks would be inevitably affected. Based on this, the reason for abandoning assigning the weights to the training tasks is not enough.
3. Some other methods in context-based meta-RL also adopt an informatic-theory-based approach to improve the generalization performance, like [1]. Though these methods are orthogonal, properly discussing them would be nice.
4. I wonder if the number of tasks will have a greater impact on the performance of the proposed method.
5. Solving the proposed algorithm needs to optimize two bi-level problems iteratively. It seems that there may exist some instability. Do the authors use some tricks to stabilize the training process?

[1] Towards an Information Theoretic Framework of Context-Based Offline Meta-Reinforcement Learning. Lanqing Li et al.

**Questions:**

1. There may be a typo in Appendix D (DERIVATION OF THE CONDITIONAL MUTUAL INFORMATION). In the equation (b), where is the remaining term $P(\\{\mathcal{T}^{cri}\_{i} \\}^{N^{cri}}\_{i=1})$ behind the term $P(\lambda)$.
2. In eq. (5), why the posterior distribution $P^*(\cdot|  \\{\overline{\mathcal{T}}^{cri}\_{i}(\lambda_i) \\}^{N^{cri}}\_{i=1}  )$ can be equal to a maximizing problem and why $P^*(\cdot|  \\{{\mathcal{T}}^{cri}\_{i}(\lambda_i) \\}^{N^{cri}}\_{i=1}  )$ is equal to the expectation of $P^*(\cdot|  \\{\overline{\mathcal{T}}^{cri}\_{i}(\lambda_i) \\}^{N^{cri}}\_{i=1}  )$ ?
3. Can the same conclusion be transferred to the offline meta-RL setting directly?

---

### Official Review · Reviewer_E7qF · 2024-10-18

**Soundness:** 3
**Presentation:** 3
**Contribution:** 3
**Rating:** 6
**Confidence:** 2

**Summary:**

This paper addresses generalization in meta-RL. To do so, the paper proposes to rank the tasks most critical to performance on the most difficult tasks, using bi-level optimization. Rather than relying on these weights alone for meta-training, which can bias the distribution to favor the critical tasks, the authors aim to improve the mutual information between the parameters and the task augmentation, given the critical tasks. The authors conduct analysis to show that this makes the model focus more on the critical tasks and that it improves generalization over the entire distribution. The authors give convergence guarantees an evaluate on MuJoCo, stock market data, and a real-world drone task.

**Strengths:**

The strengths of this paper are its command of related work, its extensive analysis, and the inclusion of real world experiments on drones. The paper clearly addresses an important issue of generalization in meta-RL. It is unclear how much more usable this method is compared to other task augmentation methods, but it appears to yield improved performance given the experiments, and it seems to address the question of how to optimally augment the task distribution.

**Weaknesses:**

The greatest weakness of this paper are its presentation and significance. While the question of optimal augmentation in meta-RL is academically interesting, it is unclear how great of an impact this will have on the field at large. However, I think some researchers will be interested, and I think it is of sufficient relevance to be considered for publication. The presentation could use work as well. More explanation of intuitions, fewer references to long proofs in the appendix, easier to parse notation, etc., would go a long way. Perhaps in the algorithm, instead of just referencing external equations, there might be a simplified example or at least an objective that could be mentioned to convey what is going on? Finally, the results appear with little discussion and and could be presented better, but are surprisingly expansive (including real world results) for a theoretical paper. It would still be great if the paper could be evaluated on distributions broader than MuJoCo and goal navigation (on drones). I am unclear if the stock benchmark fulfills this requirement. A task like Meta-World would be sufficient.

Minor:
There are a number of parenthetical citations that should not be parenthetical, e.g., line 110.

**Questions:**

1. If you are adding an additional level of optimization to meta-learning, which already typically involves bi-level optimization, this seems like it would add significant computation. Can you give a sense of the added computation needed?
2. In addition to quantifying the additional computation, could you give a sense of how this would compare to other methods making use of additional compute? Perhaps you could apply a single-task long-horizon style PPG algorithm (Beck et al., 2023) off-the-shelf in the outer-loop to a meta-learning algorithm?
3. If you have to compute weights such that the model performs better on a held-out set of tasks, can you not just use the held-out set of tasks to further optimize the meta-learning in an additional outer-loop? For example, find the best initialization such that when you meta-learn on it, you're meta-learning performance increases on the held-out set? (Or something along those lines.)
4. Can you speak to the breadth of the task distributions evaluated? They seem fairly narrow.
5. Could you give an intuition for why it is okay to "focus more on the critical tasks" and how this does not encounter the bias induced by re-weighting to focus on those tasks?

---

### Official Review · Reviewer_RpZ9 · 2024-10-29

**Soundness:** 2
**Presentation:** 2
**Contribution:** 2
**Rating:** 3
**Confidence:** 4

**Summary:**

This paper aims to improve generalization of meta-RL via understanding why meta-RL did not do well in some tasks.
The proposed  methodology has two parts: The first part identifies “critical” training tasks that are most important to achieve good performance on those poorly-adapted tasks; the second part formulates a bi-level optimization problem where
the upper level learns how to use data augmentation so that the meta-prior gives higher weights to morecritical tasks, and the lower level computes the meta-prior distribution corresponding to the current augmentation.

**Strengths:**

This paper aims to first understand why meta-RL could not generalize well in some tasks, based on which it proposed a bi-level  optimization approach to  improve generalization of meta-RL.

**Weaknesses:**

1. This paper considered meta-RL, but there is nothing special about RL in the method design. A key ingredient of RL is that  the RL agent would interact with the environment to generate new samples; this is different from supervised learning. Nevertheless,  the proposed design method can be directly used in supervised meta-learning.  The main technical component is just to use bilevel optimization to find the best coefficient in previous developed mixup data augmentation. This can be done in any learning scenarios. I suggest that the authors clarify

2. The mapping from state-action space to reward is nonlinear in general, indicating  data mixture in the proposed data augmentation would not be valid samples.

3. The knowledge of poorly-adapted validation tasks may not be available; focusing more on these poorly-adapted tasks could impede learning of other tasks, and augmenting critical tasks for these poorly-adapted tasks does not conceptually differ too much from using a larger weight, which could still not be able to improve the overall generalization performance. I suggest that the authors look into these issues further.

**Questions:**

Are you considering offline RL or online RL?

  The observation “poor generalization is that the meta-prior does not pay enough attention to the critical training tasks” is well known,  which is however claimed as a main contribution.  why?

---

### Official Review · Reviewer_oeu8 · 2024-11-03

**Soundness:** 2
**Presentation:** 1
**Contribution:** 1
**Rating:** 3
**Confidence:** 4

**Summary:**

This paper proposes an explainable method for meta-RL. This works by identifying the most challenging tasks in the task distribution, augmenting only their data, then training the policy on the augmented data from hard tasks and original data from the remaining tasks. The paper presents theoretical analysis proving algorithm convergence under Lipschitz-continuity constraints and attempts to show generalization guarantees. Experiments are performed on a physical drone navigation task and two simulated environments (stock market and MuJoCo), with comparisons to three baselines with MAML as the core meta-RL algorithm.

**Strengths:**

* The paper's contributions are clearly presented in the introduction.
* The theoretical convergence guarantees are sound.
* A thorough review of related work in explainable ML is presented.

**Weaknesses:**

## Vague and Unsupported Claims
A large number of vague, unsupported, or false claims are made throughout the work. This is worsened by the paper's writing and English being poor at many points. To list a few examples:
* "We propose the first explainable meta-RL method" - This is vague since "explainable" is not clearly defined. Many prior task inference methods could be called explainable, as the inferred latent distribution can be used to generate interpretable predictions about the environment [1].
* "Since this new meta-policy generalizes well to additional tasks compared to the original meta-policy...the generalization over the whole task distribution is likely to improve." This is provably false, as it violates core No Free Lunch theorems in RL.
* "The proposed task augmentation...does not compromise the performance on the non-critical tasks" - this is unsupported theoretically or empirically.
* "The meta-prior trained on the augmented data stores more information of the critical task" - the augmentation method (linearly interpolating state-action-advantage triplets) does not add information about the target task, it just injects randomness into the data so increases entropy. This "additional information" is noise and it is unclear why this would improve performance.
* "Show that our method outperforms state-of-the-art baselines" - the core meta-RL algorithm, MAML, is far from the state of the art and has been surpassed by a number of works in black-box meta-RL [1, 3].

In addition to this, there are multiple cases of algorithms being anthropomorphised, such as them "providing an explanation" and "paying attention to some important tasks", which are not rigorous terms and do not help in understanding the method.

## Method Ambiguity
After reading the work, it is unclear why the data augmentation method will improve generalization performance. As far as I can tell, the augmentation method applies linear interpolation between state-action-advantage triplets in the dataset, but the resulting values would have no reason to be valid in the source environment. Furthermore, the idea that this "adds information" to the task is false as the "new data"  is just noise. Finally, there's a repeated assumption that augmenting the data on challenging tasks will not impact performance on the remaining tasks, yet somehow improve performance on the challenging tasks. By No Free Lunch Theorems, this is cannot be the case.

## Missing Related Work
A large amount of prior work is omitted from the related work section. Namely, two highly relevant fields are omitted entirely: black-box meta-RL and unsupervised environment design (UED). The first of these aims to solve the same problem by learning policies with full memory across episodes [1, 3, 4], or parameterized objective functions to update agents [2, 5, 6]. UED [7, 8] studies the automatic generation of training environment distributions in order to maximise generalization performance. A common objective in this setting is minimax-regret, which makes the objective of the policy to maximise performance on the hardest training task. This is highly similar to the objective proposed in this work, yet it is uncited. The most relevant method to this paper is [2], which applies UED to meta-RL to learn a general-purpose objective function. Discussion of how this work compares to prior work from each of these fields would strengthen the contribution significantly.

[1] L. Zintgraf, K. Shiarlis, M. Igl, S. Schulze, Y. Gal, K. Hofmann, and S. Whiteson. Varibad: a very good method for bayes-adaptive deep rl via meta-learning. Proceedings of ICLR 2020, 2020.

[2] Matthew Thomas Jackson, Minqi Jiang, Jack Parker-Holder, Risto Vuorio, Chris Lu, Gregory Farquhar, Shimon Whiteson, and Jakob Nicolaus Foerster. Discovering general reinforcement learning algorithms with adversarial environment design. In Thirty-seventh Conference on Neural Information Processing Systems, 2023.

[3] Kate Rakelly, Aurick Zhou, Deirdre Quillen, Chelsea Finn, Sergey Levine. Efficient Off-Policy Meta-Reinforcement Learning via Probabilistic Context Variables. arXiv, 2019.

[4] Y. Duan, J. Schulman, X. Chen, P. L. Bartlett, I. Sutskever, and P. Abbeel. Rl 2: Fast reinforcement
learning via slow reinforcement learning. arXiv preprint arXiv:1611.02779, 2016.

[5] Matthew Jackson, Chris Lu, Louis Kirsch, Robert Lange, Shimon Whiteson, and Jakob Foerster. Discovering temporally-aware reinforcement learning algorithms. ICLR 2024.

[6] Junhyuk Oh, Matteo Hessel, Wojciech M Czarnecki, Zhongwen Xu, Hado van Hasselt, Satinder Singh, and David Silver. Discovering reinforcement learning algorithms. arXiv preprint arXiv:2007.08794, 2020.

[7] M. Jiang, M. Dennis, J. Parker-Holder, J. Foerster, E. Grefenstette, and T. Rocktäschel. Replay-guided adversarial environment design. Advances in Neural Information Processing Systems, 34: 1884–1897, 2021.

[8] M. Jiang, E. Grefenstette, and T. Rocktäschel. Prioritized level replay. In International Conference
on Machine Learning, pages 4940–4950. PMLR, 2021.

**Questions:**

1. How does your method compare those presented in prior meta-RL and UED literature?
2. How does the augmentation method guarantee that augmented samples will be valid and useful?
3. How do you ensure augmentation will not hurt performance on the remaining tasks?

---

### Official Review · Reviewer_NHwS · 2024-11-03

**Soundness:** 3
**Presentation:** 1
**Contribution:** 3
**Rating:** 6
**Confidence:** 4

**Summary:**

The paper proposes a meta-learning method to improve generalization in RL. The key observation of the paper is that poor generalization stems from poor adaptation of the meta-prior to certain (critical) tasks. Building on this observation, the paper proposes first to identify the critical tasks by learning a weight vector that scores the importance of the different tasks. Next, to solve a bi-level optimization method, where the upper level learns an optimal augmentation for the critical tasks (optimal in the sense that the mutual information between the meta-parameter and the augmented critical tasks is maximized), and the lower level learns the optimal meta-prior given the optimal augmentation learned by the upper level.
The authors theoretically prove that the algorithm converges and improves generalization. In addition, the paper demonstrates that the proposed approach improves the performance of standard meta-RL algorithms in two real-world problems and three MuJoCo experiments.

**Strengths:**

* The paper tackles the important problem of poor generalization in meta-RL.

* There is a good discussion about relevant work on explainable RL and generalization in meta-learning.

* The experiment section is impressive - it includes two real-world problems in addition to three MuJoCo experiments.

**Weaknesses:**

* The main weakness of the paper is its writing quality - many sentences are repetitive without any additional information. For example, almost every section reiterates that the key observation of the paper is to pay more attention to the critical tasks. I suggest removing all repetitive text.

* In addition, I recommend extending the experiment section with more ablation studies and explanations of the experimental setting. For example, the sim-to-real part of the drone navigation experiment seems to me quite important.

* An ablation study on Ncr (the number of critical tasks) is missing.

* Since the baselines (Task weighting, Meta augmentation, Meta regularization) were not originally tested in these specific real-world and MuJoCo experiments, it is unclear how the hyperparameter search was performed, and how their original implementations were adapted.

**Questions:**

1. Why were only 5 critical tasks used in all experiments?  Is it the optimal number of critical tasks for all problems? Since the problems differ from each other, my intuition is that their optimal number of critical tasks is also different.

2. As explained in remark 1 (lines 200-210) the weighted meta-policy learned in problem (2) can not be used to improve generalization.
Could you provide an experiment that shows that this hypothesis indeed holds?

3. How was the hyperparameter search done for all the baselines?

---

### Meta-Review · Area_Chair_2utf · 2024-12-22

**Metareview:**

The paper proposes a meta-learning method to improve generalization in RL. The key observation of the paper is that poor generalization stems from poor adaptation of the meta-prior to certain (critical) tasks. Building on this observation, the paper proposes first to identify the critical tasks by learning a weight vector that scores the importance of the different tasks.

This paper received mixed scores of 3, 3, 6, 6, with several concerns about the scope of the claims, imperfect justification, and ambiguity in understanding the method. With no author responses, this paper cannot be accepted.

**Additional Comments On Reviewer Discussion:**

No discussion due to no responses from authors.

---

### Decision · Program_Chairs · 2025-01-22

Reject